# Approximate Probabilistic Inference with Composed Flows

## Abstract

We study the problem of probabilistic inference on the joint distribution defined by a normalizing flow model. Given a pre-trained flow model $p(\boldsymbol{x})$, we wish to estimate $p(\boldsymbol{x}_2 \mid \boldsymbol{x}_1)$ for some arbitrary partitioning of the variables $\boldsymbol{x} = (\boldsymbol{x}_1, \boldsymbol{x}_2)$. We first show that this task is computationally hard for a large class of flow models. Motivated by this hardness result, we propose a framework for *approximate* probabilistic inference. Specifically, our method trains a new generative model with the property that its composition with the given model approximates the target conditional distribution. By parametrizing this new distribution as another flow model, we can efficiently train it using variational inference and also handle conditioning under arbitrary differentiable transformations. Since the resulting approximate posterior remains a flow, it offers exact likelihood evaluation, inversion, and efficient sampling. We provide an extensive empirical evidence showcasing the flexibility of our method on a variety of inference tasks with applications to inverse problems. We also experimentally demonstrate that our approach is comparable to simple MCMC baselines in terms of sample quality. Further, we explain the failure of naively applying variational inference and show that our method does not suffer from the same issue.

## 1 Introduction

Generative modeling has seen an unprecedented growth in the recent years. Building on the success of deep learning, deep generative models have shown impressive ability to model complex distributions in a variety of domains and modalities. Among them, normalizing flow models (see Papamakarios et al. (2019) and references therein) stand out due to their computational flexibility, as they offer efficient sampling, likelihood evaluation, and inversion. While other types of models currently outperform flow models in terms of likelihood and sample quality, flow models have the advantage that they are relatively easy to train using maximum likelihood and do not suffer from issues that other models possess (e.g. mode collapse for GANs, posterior collapse for VAEs, slow sampling for autoregressive models). These characteristics make normalizing flows attractive for a variety of downstream tasks, including density estimation, inverse problems, semi-supervised learning, reinforcement learning, and audio synthesis (Ho et al., 2019; Asim et al., 2019; Atanov et al., 2019; Ward et al., 2019; Oord et al., 2018).

Even with such computational flexibility, how to perform efficient probabilistic inference on a flow model still remains largely unknown. This question is becoming increasingly important as generative models increase in size and the computational resources necessary to train them from scratch are out of reach for many researchers and practitioners[1]. If it was possible to perform probabilistic inference on flow models, we could *re-purpose* these powerful pre-trained generators for numerous custom tasks.

This is the central question we study in this paper: One wishes to estimate the conditional distribution $p(\boldsymbol{x}_2 \mid \boldsymbol{x}_1)$ from a given flow model $p(\boldsymbol{x})$ for some partitioning of variables $\boldsymbol{x} = (\boldsymbol{x}_1, \boldsymbol{x}_2)$. Existing methods for this task largely fall under two categories: Markov Chain Monte Carlo (MCMC) and variational inference (VI). While MCMC methods can perform exact conditional sampling in theory, they often have prohibitively long mixing time for complex high-dimensional distributions and also

---

[1]For example, Kingma & Dhariwal (2018) report that their largest model had 200M parameters and was trained on 40 GPUs for a week.

do not provide likelihoods. On the other hand, VI allows for approximate likelihood evaluation under the variational posterior and fast sampling, but at a lower sample quality compared to MCMC counterparts.

We propose a novel method that leverages a powerful pre-trained flow model *by constructing carefully designed latent codes* to generate conditional samples via variational inference. While this procedure is intractable for latent variable models in general, the invertibility of the pre-trained model gives us a tractable algorithm for learning a distribution in the latent space whose samples approximately match the true conditional when fed into the pre-trained model.

**Our contributions:**

- We start with an interesting theoretical hardness result. We show that even though flow models are designed to provide efficient inversion and sampling, even *approximate* sampling from the exact conditional distribution is provably computationally intractable for a wide class of flow models. This motivates our approach of smoothing the observation.

- We develop a method to estimate the target conditional distribution by composing a second flow model (which we call the *pre-generator*) with the given model. In particular, this parametrization allows us to employ variational inference and avoid unstable adversarial training as was explored in existing work.

- The resulting approximate posterior retains the computational flexibility of a flow model and can be used on a wide variety of downstream tasks that require fast sampling, exact likelihood evaluation, or inversion. Compared to MCMC methods, it has the benefit of being able to generate samples that are guaranteed to be i.i.d.

- We experimentally show that our approach is comparable to simple MCMC baselines in terms of sample quality metrics such as Frechet Inception Distance (Heusel et al., 2017). We also demonstrate that it achieves superior conditional likelihood estimation performance compared to regular variational inference.

- We extend and validate our method for conditioning under arbitrary differentiable transformations with applications to inverse problems. We qualitatively demonstrate its flexibility on various complex inference tasks.

## 2 BACKGROUND

### 2.1 NORMALIZING FLOWS

Normalizing flow models (also known as invertible generative models) represent complex probability distributions by transforming a simple input noise $z$ (typically standard Gaussian) through a differentiable bijection $f : \mathbb{R}^d \to \mathbb{R}^d$. Since $f$ is invertible, change of variables formula allows us to compute the probability density of $x = f(z)$:

$$\log p(\boldsymbol{x}) = \log p(\boldsymbol{z}) + \log \left| \det \frac{df^{-1}}{d\boldsymbol{x}}(\boldsymbol{x}) \right|,$$

where $\frac{df^{-1}}{d\boldsymbol{x}}$ denotes the Jacobian of the inverse transformation $f^{-1} : \boldsymbol{x} \mapsto \boldsymbol{z}$. Flow models are explicitly designed so that the above expression can be easily computed, including the log-determinant term. This tractability allows them to be directly trained with maximum likelihood objective on data.

Starting from the early works of Dinh et al. (2015) and Rezende & Mohamed (2015), there has been extensive research on invertible architectures for generative modeling. Many of them work by composing a series of invertible layers, such as in RealNVP (Dinh et al., 2016), IAF (Kingma et al., 2016), Glow (Kingma & Dhariwal, 2018), invertible ResNet (Behrmann et al., 2019), and Neural Spline Flows (Durkan et al., 2019).

One of the simplest invertible layer construction is *additive coupling layer* introduced by Dinh et al. (2015), which served as a basis for many other subsequently proposed models mentioned above. In an additive coupling layer, the input variable is partitioned as $\boldsymbol{x} = (\boldsymbol{x}_1, \boldsymbol{x}_2) \in \mathbb{R}^{d_1} \times \mathbb{R}^{d_2}$. The layer is parametrized by a neural network $g(\boldsymbol{x}_1) : \mathbb{R}^{d_1} \to \mathbb{R}^{d_2}$ used to additively transform $\boldsymbol{x}_2$. Thus the layer's output $\boldsymbol{y} = (\boldsymbol{y}_1, \boldsymbol{y}_2) \in \mathbb{R}^{d_1} \times \mathbb{R}^{d_2}$ and its inverse can be computed as follows:

$$\begin{cases} \boldsymbol{y}_1 &= \boldsymbol{x}_1 \\ \boldsymbol{y}_2 &= \boldsymbol{x}_2 + g(\boldsymbol{x}_1) \end{cases} \iff \begin{cases} \boldsymbol{x}_1 &= \boldsymbol{y}_1 \\ \boldsymbol{x}_2 &= \boldsymbol{y}_2 - g(\boldsymbol{y}_1) \end{cases}$$

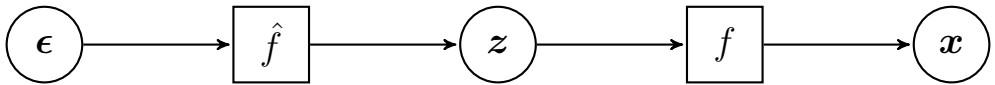

Figure 1: A flow chart of our conditional sampler. First the noise variable $\epsilon$ is sampled from $\mathcal{N}(\mathbf{0}, I_d)$. This is fed into our pre-generator $\hat{f}$ to output structured noise $z$, which is driving the pre-trained base model $f$ to generate conditional samples $x$. The central idea is that the pre-generator must produce structured noise $z$ that looks Gaussian (so that the samples are realistic) but also make the pre-trained base model $f$ produce samples that satisfy the conditioning. The final conditional sampler is thus defined by the composition of two flow models.

Notably, the determinant of the Jacobian of this transformation is always 1 for any mapping $g$.

## 2.2 VARIATIONAL INFERENCE

Variational inference (VI) is a family of techniques for estimating the conditional distribution of unobserved variables given observed ones (Jordan et al., 1999; Wainwright et al., 2008; Blei et al., 2017). At its core, VI tries to find a tractable approximation of the true conditional density by minimizing the KL divergence between them over a set of distributions called the variational family.

Within our context of probabilistic inference from a joint distribution $p(\boldsymbol{x}_1, \boldsymbol{x}_2)$, the KL minimization problem we solve is the following:

$$\arg\min_{q \in \mathcal{Q}} D_{\mathrm{KL}}(q(\boldsymbol{x}_2) \,\|\, p(\boldsymbol{x}_2 \mid \boldsymbol{x}_1 = \boldsymbol{x}_1^*)),$$

where $\boldsymbol{x}_1^*$ is some fixed observation and the variational family $\mathcal{Q}$ is chosen such that efficient sampling and likelihood evaluation are possible for all $q \in \mathcal{Q}$. Note that $q$ is tied to a particular $\boldsymbol{x}_1^*$ and is not shared across different observations.

While the conditional density $p(\boldsymbol{x}_2 \mid \boldsymbol{x}_1 = \boldsymbol{x}_1^*)$ is generally intractable to compute, we can efficiently evaluate the joint density $p(\boldsymbol{x}_1, \boldsymbol{x}_2)$ when it is given as a flow model. Thus, we instead perform variational inference on the conditional *joint* distribution:

$$\min_{q \in \mathcal{Q}} D_{\mathrm{KL}}(q(\boldsymbol{x}_2) \,\|\, p(\boldsymbol{x}_2 \mid \boldsymbol{x}_1 = \boldsymbol{x}_1^*)) = \min_{q \in \mathcal{Q}} \mathbb{E}_{\boldsymbol{x}_2 \sim q}[\log q(\boldsymbol{x}_2) - \log p(\boldsymbol{x}_1 = \boldsymbol{x}_1^*, \boldsymbol{x}_2)],$$

which is obtained by dropping the $\log p(\boldsymbol{x}_1 = \boldsymbol{x}_1^*)$ term that is constant w.r.t. $q$. We note that this procedure is intractable for other types of latent variable models such as VAE or GAN.

## 3 HARDNESS OF CONDITIONAL SAMPLING ON FLOWS

Before we present our method, we first establish a hardness result for exact conditional sampling for flow models that use additive coupling layer. Specifically, if an algorithm is able to efficiently sample from the conditional distribution of a flow model with additive coupling layers, then this algorithm can be used to solve NP-complete problems efficiently.

**Theorem 1.** *(Informal) Suppose there is an efficient algorithm that can draw samples from the conditional distribution $p(\boldsymbol{x}_2 \mid \boldsymbol{x}_1)$ of a normalizing flow model $p(\boldsymbol{x}_1, \boldsymbol{x}_2)$ implemented with additive coupling layers as defined in Dinh et al. (2015). Then $RP = NP$.*

Moreover, even *approximating* the true conditional distribution remains hard as long as we require that the conditioning is exact. The formal statement of the above theorem and its corollary to the approximate case can be found in Appendix A.

Importantly, this result implies that allowing *approximate sampling* does not affect the hardness of the problem, as long as the hard constraint of exact conditioning ($\boldsymbol{x}_1 = \boldsymbol{x}_1^*$) is there. Thus we are motivated to instead consider *approximate conditioning* where the conditioned variable $\boldsymbol{x}_1$ is not required to match the given observation exactly. We also note that the class of flow models that include additive coupling layers encompasses a large number of existing models (e.g. most of the models in Section 2.1). Thus our hardness result applies to a variety of flow models used in practice.

## 4 APPROXIMATE PROBABILISTIC INFERENCE WITH COMPOSED FLOWS

**Notation.** Let $p_f(\boldsymbol{x})$ denote the *base model*, a fixed flow model defined by the invertible mapping $f : \boldsymbol{z} \mapsto \boldsymbol{x}$. The *pre-generator* $p_{\hat{f}}(\boldsymbol{z})$ is similarly defined by the invertible mapping $\hat{f} : \boldsymbol{\epsilon} \mapsto \boldsymbol{z}$ and represents a distribution in the latent space of the base model. We assume that all flow models use standard Gaussian prior, i.e. $\boldsymbol{z} \sim \mathcal{N}(\mathbf{0}, I_d) \to \boldsymbol{x} = f(\boldsymbol{z})$ and $\boldsymbol{\epsilon} \sim \mathcal{N}(\mathbf{0}, I_d) \to \boldsymbol{z} = \hat{f}(\boldsymbol{\epsilon})$. By composing the base model and the pre-generator, we obtain the *composed model* $p_{f \circ \hat{f}}(\boldsymbol{x})$ whose samples are generated via $\boldsymbol{\epsilon} \sim \mathcal{N}(\mathbf{0}, I_d) \to \boldsymbol{x} = f(\hat{f}(\boldsymbol{\epsilon}))$. Figure 1 details this sampling procedure.

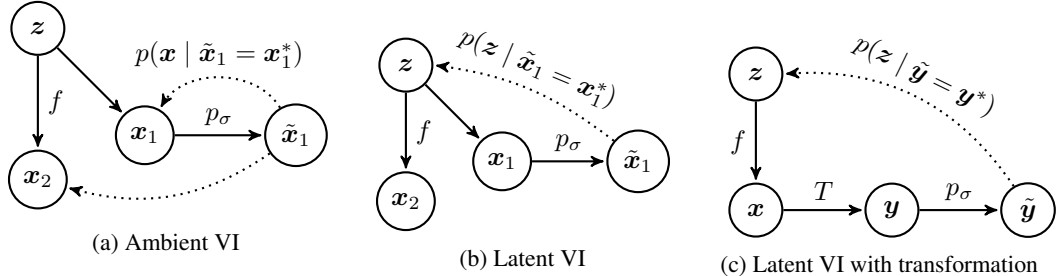

Figure 2: Graphical models depicting different ways we perform variational inference. Solid arrows represent the generative direction, and dotted arrows indicate the variational direction.

**Our method.** Following the principle of leveraging the favorable geometry of the latent space commonly employed in VI and MCMC literature, we also perform our inference in the latent space of the base model. Because the base model maps Gaussian noise to samples, the pre-generator can simply focus on putting probability mass in the regions of the latent space corresponding to the conditional samples we wish to model. This is in contrast with regular variational inference, which essentially trains an entirely new generative model from scratch to approximate the target conditional density.

Motivated by this observation and the hardness of conditional sampling, we propose to perform variational inference *in the latent space on a smoothed version of the problem* with Gaussian smoothing. Specifically, we create a smoothed variable $\tilde{\boldsymbol{x}}_1$ distributed according to $p_\sigma(\tilde{\boldsymbol{x}}_1 \mid \boldsymbol{x}_1) = \mathcal{N}(\tilde{\boldsymbol{x}}_1; \boldsymbol{x}_1, \sigma^2 I_d)$ and condition on $\tilde{\boldsymbol{x}}_1 = \boldsymbol{x}_1^*$ instead. Here, $\sigma$ is the *smoothing parameter* that controls the tightness of this relaxation. We have experimented with non-Gaussian smoothing kernels such as the one implicitly defined by the perceptual distance measure LPIPS (Zhang et al., 2018). While we chose to use Gaussian smoothing because our preliminary results showed no appreciable difference, we note that a more extensive tuning of the smoothing kernel may lead to a better sample quality, which we leave for future work.

Thus the objective we would like to minimize is the KL divergence between our variational distribution and the smoothed conditional density:

$$D_{\mathrm{KL}}(p_{f \circ \hat{f}}(\boldsymbol{x}_2) \,\|\, p_f(\boldsymbol{x}_2 \mid \tilde{\boldsymbol{x}}_1 = \boldsymbol{x}_1^*)). \tag{1}$$

For comparison, a direct application of variational inference in the image space (which we refer to as *Ambient VI*) would minimize the following objective:

$$D_{\mathrm{KL}}(p_g(\boldsymbol{x}_2) \,\|\, p_f(\boldsymbol{x}_2 \mid \tilde{\boldsymbol{x}}_1 = \boldsymbol{x}_1^*)), \tag{2}$$

where we write $p_g$ to denote the variational distribution that directly models $f(\boldsymbol{x} \mid \tilde{\boldsymbol{y}} = \boldsymbol{y}^*)$ in the ambient (image) space, parametrized by an invertible mapping $g : \boldsymbol{z} \mapsto \boldsymbol{x}$.

**Intractability of equation 1.** Note that in our setting, we cannot directly approximate $p_f(\boldsymbol{x}_2 \mid \tilde{\boldsymbol{x}}_1 = \boldsymbol{x}_1^*)$ because we only have access to the joint distribution $p_f(\boldsymbol{x}_1, \boldsymbol{x}_2)$ through the base model. Fortunately, the VI loss for approximating the *joint* conditional $p_f(\boldsymbol{x} \mid \tilde{\boldsymbol{x}}_1 = \boldsymbol{x}_1^*)$ is an upper bound to eq. (1):

$$D_{\mathrm{KL}}(p_{f \circ \hat{f}}(\boldsymbol{x}) \,\|\, p_f(\boldsymbol{x} \mid \tilde{\boldsymbol{x}}_1 = \boldsymbol{x}_1^*)) \geq D_{\mathrm{KL}}(p_{f \circ \hat{f}}(\boldsymbol{x}_2) \,\|\, p_f(\boldsymbol{x}_2 \mid \tilde{\boldsymbol{x}}_1 = \boldsymbol{x}_1^*)),$$

which we prove in Appendix B.2. Thus we are justified in our use of the joint VI loss instead of eq. (1). Moreover, this concern vanishes for the general case where we observe a transformation of $\boldsymbol{x}$, as discussed below. Figures 2a and 2b show the graphical models reflecting this formulation.

**Conditioning under differentiable transformations.** The flexibility of VI allows us to easily extend our method to conditioning under an arbitrary transformation. Concretely, let $T(\boldsymbol{x})$ be a differentiable function and $\boldsymbol{y}^*$ be some fixed observation in the range of $T$. We now observe $\boldsymbol{y} = T(\boldsymbol{x})$ instead, so we similarly define a smoothed variable $\tilde{\boldsymbol{y}}$ distributed according to $p_\sigma(\tilde{\boldsymbol{y}} \mid \boldsymbol{y}) = \mathcal{N}(\tilde{\boldsymbol{y}}; \boldsymbol{y}, \sigma^2 I_d)$. We estimate the conditional density $p_f(\boldsymbol{x} \mid \tilde{\boldsymbol{y}} = \boldsymbol{y}^*)$ by minimizing the following objective:

$$\begin{aligned}
\mathcal{L}_{\mathrm{Ours}}(\hat{f}) &\triangleq D_{\mathrm{KL}}(p_{f \circ \hat{f}}(\boldsymbol{x}) \,\|\, p_f(\boldsymbol{x} \mid \tilde{\boldsymbol{y}} = \boldsymbol{y}^*)) \\
&= D_{\mathrm{KL}}(p_{\hat{f}}(\boldsymbol{z}) \,\|\, p_f(\boldsymbol{z})) + \mathbb{E}_{\boldsymbol{z} \sim p_{\hat{f}}}\left[\frac{1}{2\sigma^2} \|T(f(\boldsymbol{z})) - \boldsymbol{y}^*\|_2^2\right],
\end{aligned} \tag{3}$$

where $p_f(\boldsymbol{z})$ denotes the prior distribution of the base model, i.e. $\mathcal{N}(\boldsymbol{0}, I_d)$. We provide the derivation of eq. (3) in Appendix B.1. See Figure 2c for a comparison to Equation (1).

This loss function offers an intuitive interpretation. The first term tries to keep the learned distribution $p_{f \circ \hat{f}}$ close to the base distribution by pushing $p_{\hat{f}}$ to match the prior of the base model, while the second term tries to match the observation $\boldsymbol{y}^*$. Moreover, the above expectation can be rewritten in terms of $\boldsymbol{\epsilon}$, which allows us to employ the *reparametrization trick* to obtain a low-variance gradient estimator for training:

$$
\begin{aligned}
\mathcal{L}_{\text{Ours}}(\hat{f}) &= \mathbb{E}_{\boldsymbol{z} \sim p_{\hat{f}}} \left[ \log p_{\hat{f}}(\boldsymbol{z}) - \log p_f(\boldsymbol{z}) + \frac{1}{2\sigma^2} \left\| T(f(\boldsymbol{z})) - \boldsymbol{y}^* \right\|_2^2 \right] \\
&= \mathbb{E}_{\boldsymbol{\epsilon} \sim \mathcal{N}(\boldsymbol{0}, I_d)} \left[ \log p_{\hat{f}}(\hat{f}(\boldsymbol{\epsilon})) - \log p_f(\hat{f}(\boldsymbol{\epsilon})) + \frac{1}{2\sigma^2} \left\| T(f(\hat{f}(\boldsymbol{\epsilon}))) - \boldsymbol{y}^* \right\|_2^2 \right]
\end{aligned}
\tag{4}
$$

## 5 RELATED WORK

**Conditional Generative Models.** There has been a large amount of work on conditional generative modeling, with varying levels of flexibility for what can be conditioned on. In the simplest case, a fixed set of observed variables can be directly fed into the model as an auxiliary conditioning input (Mirza & Osindero, 2014; Sohn et al., 2015; Ardizzone et al., 2019). Some recent works proposed to extend existing models to support conditioning on *arbitrary* subsets of variables (Ivanov et al., 2018; Belghazi et al., 2019; Li et al., 2019). This is a much harder task as there are exponentially many subsets of variables that can be conditioned on.

More relevant to our setting is (Engel et al., 2017), which studied conditional sampling from *non-invertible* latent variable generators such as VAE and GAN. It proposes to adversarially train a GAN-style generator within the latent space of a pre-trained latent variable model, thereby avoiding the issue of intractability of variational inference for non-invertible models. Due to adversarial training and the lack of invertibility of the base model, however, the learned conditional sampler loses the computational flexibility of our method.

We highlight several reasons why one might prefer our approach over the above methods: (1) The data used to train the given model may not be available, and only the generative model itself is made public. (2) The given model is too costly to train from scratch. (3) We wish to perform custom downstream tasks (e.g. compression, inversion) that are difficult to do with other parametrizations of the posterior (4) We need to condition on a transformation of the variables, instead of just a subset of them. (5) We want to get some insight on the distribution defined by the given model.

**Markov Chain Monte Carlo Methods.** When one is only concerned with generating samples, MCMC techniques offer a promising alternative. Unlike variational inference and conditional models, MCMC methods come with asymptotic guarantees to generate correct samples. Though directly applying MCMC methods on complex high-dimensional posteriors parametrized by a neural network often comes with many challenges in practice (Papamarkou et al., 2019), methods based on Langevin Monte Carlo have shown promising results (Neal et al., 2011; Welling & Teh, 2011; Song & Ermon, 2019). Moreover, recent works by Parno & Marzouk (2018) and Hoffman et al. (2019) also leveraged the favorable geometry of the latent space of a flow model to improve mixing of MCMC chains. This idea was later adapted by Nijkamp et al. (2020) in the context of training energy-based models.

While our VI-based method lacks asymptotic guarantees, we hope to show that it is empirically competitive with simple MCMC techniques. For this purpose, we consider two baselines: Langevin dynamics and PL-MCMC (Cannella et al., 2020). PL-MCMC is particularly relevant as it also tackles the task of conditional sampling from a flow model. Unlike our method, it constructs a Markov chain in the latent space and uses Metropolis-Hastings to sample from the target conditional distribution with asymptotic guarantees.

**Inverse Problems with Generative Prior.** In a linear inverse problem, a vector $\boldsymbol{x} \in \mathbb{R}^d$ generates a set of measurements $\boldsymbol{y}^* = A\boldsymbol{x} \in \mathbb{R}^m$, where the number of measurements is much smaller than the dimension: $m \ll d$. The goal is to reconstruct the vector $\boldsymbol{x}$ from $\boldsymbol{y}^*$. While in general there are (potentially infinitely) many possible values of $\boldsymbol{x}$ that agree with the given measurements, it is possible to identify a unique solution when there is an additional structural assumption on $\boldsymbol{x}$.

Classically, the simplifying structure was that $\boldsymbol{x}$ is sparse, and there has been extensive work in this setting (Tibshirani, 1996; Candes et al., 2006; Donoho et al., 2006; Bickel et al., 2009; Baraniuk, 2007). Recent work has considered alternative structures, such as the vector $\boldsymbol{x}$ coming from a generative model. Starting with Bora et al. (2017), there has been extensive work in this setting as

well (Grover & Ermon, 2019; Mardani et al., 2018; Heckel & Hand, 2019; Mixon & Villar, 2018; Pandit et al., 2019). In particular, Asim et al. (2019) and Shamshad et al. (2019) utilized flow models to solve inverse problems such as compressive sensing, image deblurring, and image inpainting.

It is important to note that the above approaches focus on recovering a single point estimate that best matches the measurements. However, there can be many inputs that fit the measurements and thus uncertainty in the reconstruction. Due to this shortcoming, several recent works focused on recovering the *distribution* of $x$ conditioned on the measurements (Tonolini et al., 2019; Zhang & Jin, 2019; Adler & Öktem, 2018; 2019). We note that our approach differs from these, since they are learning-based methods that require access to the training data. On the contrary, our work attempts to perform conditional sampling using a given pre-trained generative model, leveraging all that previous computation to solve a conditional task with reconstruction diversity.

## 6 QUANTITATIVE EXPERIMENTS

We validate the efficacy of our proposed method in terms of both sample quality and likelihood on various inference tasks against three baselines: Ambient VI (as defined by the loss in Equation (2)), Langevin Dynamics, and PL-MCMC. We also conduct our experiments on three different datasets (MNIST, CIFAR-10, and CelebA-HQ) to ensure that our method works across a range of settings.

We report four different sample quality metrics: Frechet Inception Distance (FID), Learned Perceptual Image Patch Similarity (LPIPS), and Inception Score (IS) for CIFAR-10 (Heusel et al., 2017; Zhang et al., 2018; Salimans et al., 2016). While not strictly a measure of perceptual similarity, the average mean squared error (MSE) is also reported for completeness.

For all our experiments, we use the multiscale RealNVP architecture (Dinh et al., 2016) for both the base model and the pre-generator. We use Adam optimizer (Kingma & Ba, 2014) to optimize the weights of the pre-generator using the loss defined in Equation (3). The images used to generate observations were taken from the test set and were not used to train the base models. We refer the reader to Appendix C for model hyperparameters and other details of our experiment setup.

| Original | Observed | Conditional Samples | Variance |
|---|---|---|---|

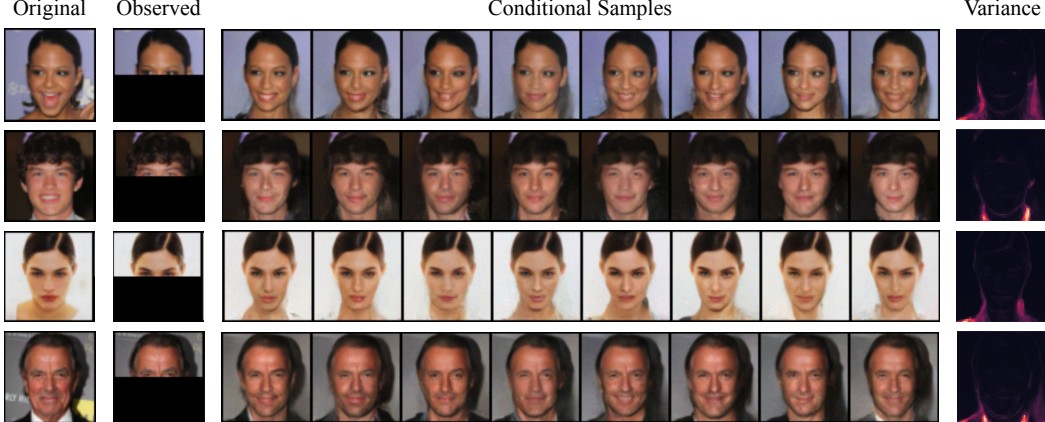

Figure 3: Conditional samples generated by our method from observing the upper half of CelebA-HQ faces. We see that our approach is able to produce diverse completions with different jaw lines, mouth shapes, and facial expression.

### 6.1 IMAGE INPAINTING

We perform inpainting tasks using our approach, where we sample missing pixels conditioned on the visible ones. We consider three different conditioning schemes: the bottom half (MNIST), the upper half (CelebA-HQ), and randomly chosen subpixels (CIFAR-10). For MNIST, we use the smoothing parameter value of $\sigma = 0.1$ and for CIFAR-10 and CelebA-HQ, we use $\sigma = 0.05$.

In Figure 3 we see that our approach produces natural and diverse samples for the missing part of the image. The empirical pixelwise variance (normalized and averaged over the color channels) also confirms that, while the observation is not perfectly matched, most of the high-variance regions are in the unobserved parts as we expect.

We also quantitatively evaluate the quality of the generated samples using widely used sample quality metrics, as shown in Table 1. As we can see, our method outperforms the baseline methods on most

of the metrics. Note that PL-MCMC results for CIFAR-10 and CelebA-HQ are omitted because it was prohibitively slow to run for hundreds of images, as each MCMC chain required over 20,000 proposals. Cannella et al. (2020) also report using 25,000 proposals for their experiments.

Table 1: Sample quality metrics for image inpainting tasks on different datasets. The best value is bolded for each metric. As shown below, our method achieves superior sample quality to all baselines.

| | MNIST | | | CIFAR-10 (5-bit) | | | | CelebA-HQ (5-bit) | | |
|---|---|---|---|---|---|---|---|---|---|---|
| | FID | MSE | LPIPS | FID | IS↑ | MSE | LPIPS | FID | MSE | LPIPS |
| Ours | **4.11** | **21.67** | **0.074** | **41.14** | **7.189** | 9.71 | **0.176** | 33.61 | **223.06** | **0.208** |
| Langevin | 14.34 | 36.51 | 0.135 | 47.53 | 6.732 | **9.31** | 0.201 | **30.33** | 323.47 | 0.229 |
| Ambient VI | 114.59 | 65.56 | 0.290 | 84.78 | 5.156 | 16.74 | 0.296 | 289.64 | 1060.66 | 0.587 |
| PL-MCMC | 21.20 | 59.89 | 0.190 | | N/A | | | | N/A | |

## 6.2 LIKELIHOOD ESTIMATION

Next, we evaluate our method on the task of conditional likelihood estimation. By varying the size of the pre-generator, we also test the parameter efficiency of our method in comparison to Ambient VI. Results are shown in Table 2; we see that our method is able to produce reasonable samples using only *about 1%* of the base model's parameters, confirming the effectiveness of inference in the latent space.

Table 2: Conditional likelihood estimation performance (measured in bits per dimension) for different pre-generator sizes on the MNIST imputation task. The first row shows the parameter count of the pre-generator relative to the base model.

| | Observations | | |
|---|---|---|---|
| # Parameters | Ours | Ambient VI | Conditional Completions (ours) |
| 1.2% | 1.73 | 6.75 | |
| 3.2% | 1.64 | 3.17 | |
| 10.6% | 1.52 | 2.71 | |
| 39.1% | 1.47 | 2.99 | |

## 7 QUALITATIVE EXPERIMENTS

**Extracting Class-conditional Models:** Here we present an interesting application of conditioning under a complex transformation $T$. Since the only requirement for $T$ is that it must be differentiable, we can set it to be another neural network. For example, if $T$ is a pre-trained binary classifier for a specific attribute, we can *extract* a model conditioned on the presence (or the absence) of that attribute from an unconditional base model.

We test this idea on the MNIST dataset. We trained 10 binary classifiers on the MNIST dataset, one for each digit $k = 0, \ldots, 9$, to predict whether the given image is $k$ or not. By setting $T$ to be each of those classifiers, we were able to extract the class-conditional model $p_f(\boldsymbol{x} \mid \text{Label}(\boldsymbol{x}) \approx k)$. See Figure 4a for samples generated from the extracted models.

**Inverse Problems:** We additionally test the applicability of our method to linear inverse problems. In Figure 5, we show the conditional samples obtained by our method on three different tasks: image colorization, super-resolution ($2\times$), and compressed sensing with 500 random Gaussian measurements (for reference, CIFAR-10 images have 3072 dimensions). We notice that the generated samples look natural, even when they do not match the original input perfectly, again showing our method's capability to generate semantically meaningful conditional samples and also provide sample diversity.

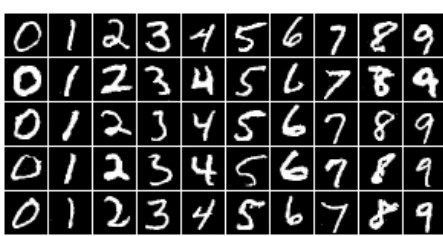

(a) Each column contains samples from the learned posterior conditioned on different MNIST classes. We emphasize that the base model was trained without class information.

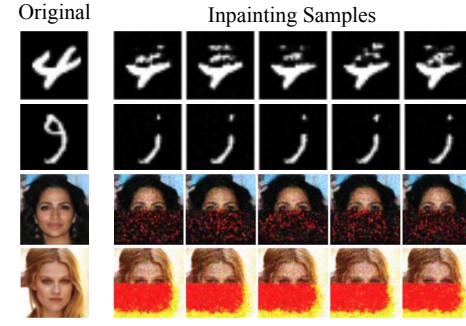

(b) Conditional samples from image inpainting experiments done with Ambient VI.

Figure 4: Samples from class-conditional models extracted from the unconditional base model (left) and various failure modes of Ambient VI (right).

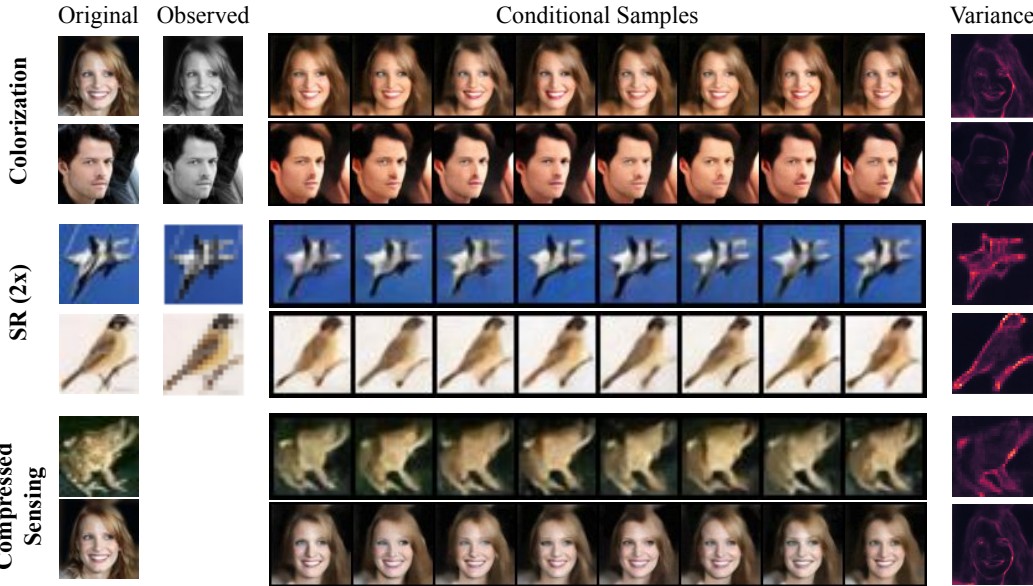

Figure 5: Results on various inverse problem tasks using our method.

### 7.1 WHY AMBIENT VI FAILS

From Table 1, notice that Ambient VI achieves significantly worse sample quality compared to other methods. The low-quality samples from the image inpainting task in Figure 4b further confirm that Ambient VI is unable to produce good conditional samples, even though the observation is matched well. This may seem initially surprising but is a natural consequence of the VI objective. Recall that our loss function decomposes into two terms: the KL term and the reconstruction term.

$$\mathcal{L}_{\text{Ours}}(\hat{f}) = D_{\text{KL}}(p_{\hat{f}}(\boldsymbol{z}) \,\|\, p_f(\boldsymbol{z})) + \mathbb{E}_{\boldsymbol{z} \sim p_{\hat{f}}} \left[ \frac{1}{2\sigma^2} \, \|T(f(\boldsymbol{z})) - \boldsymbol{y}^*\|_2^2 \right]. \qquad (5)$$

If we alternatively derive the loss for Ambient VI, we arrive at an analogous objective:

$$\mathcal{L}_{\text{Ambient}}(g) = D_{\text{KL}}(p_g(\boldsymbol{x}) \,\|\, p_f(\boldsymbol{x})) + \mathbb{E}_{\boldsymbol{x} \sim p_g} \left[ \frac{1}{2\sigma^2} \, \|T(\boldsymbol{x}) - \boldsymbol{y}^*\|_2^2 \right]. \qquad (6)$$

While these two loss functions seem like simple reparametrizations of each other via $f$, they behave very differently during optimization due to the KL term. Notice that for both loss functions, the first term is the *reverse* KL divergence between the variational distribution and the base distribution. Because reverse KL divergence places no penalty whenever $p_{\hat{f}}$ or $p_g$ is zero regardless of $p_f$, minimizing the reverse KL is known to have a *mode-seeking* behavior where variational distribution fits a single mode of $p_f$ and ignores the rest of the support of $p_f$ (Murphy, 2013, Chapter 21.2.2).

In contrast, minimizing the forward KL has a *zero-avoiding* behavior and tries to cover all of $p_f$'s support.

For our method, this is not a problem because the prior distribution of the base model $p_f(\boldsymbol{z})$ is a standard Gaussian and hence unimodal. However, for Ambient VI, $p_f(\boldsymbol{x})$ is the base model itself and is highly multimodal. This can be empirically seen by visualizing the landscape of $\log p_f(\boldsymbol{x})$ projected onto a random 2D subspace. In Figure 6, we clearly see that $p_f(\boldsymbol{x})$ has numerous local maxima. For Ambient VI, the variational distribution collapses into one of these modes.

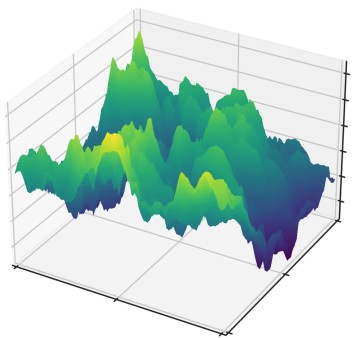

## 8 CONCLUSION

We proposed a new inference algorithm for distributions parametrized by normalizing flow models. The need for approximate inference is motivated by our theoretical hardness result for exact inference, which is surprising given that it applies to invertible models. We also presented a detailed empirical evaluation of our method with both quantitative and qualitative results on a wide range of tasks and datasets.

Figure 6: Contour plot of $\log p_f(\boldsymbol{x})$ around a random point in image space.

There are numerous directions for further research. For example, our method can be possibly extended to other latent-variable generators such as VAEs and GANs (Kingma & Welling, 2013; Goodfellow et al., 2014). A significant limitation of our work is that we need to re-train the pre-generator for each observation. It may be possible to avoid retraining by *amortizing* the pre-generator over all possible observations. Studying the various trade-offs resulting from such scheme would be an interesting result similar in spirit to Cremer et al. (2018). Overall, we believe that the idea of a pre-generator creating structured noise is a useful and general method for leveraging pre-trained generators to solve new generative problems.

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

## A   PROOF OF HARDNESS RESULTS

### A.1   PRELIMINARIES

A Boolean variable is a variable that takes a value in $\{-1, 1\}$. A *literal* is a Boolean variable $x_i$ or its negation $\neg x_i$. A *clause* is set of literals combined with the OR operator, e.g., $x_1 \vee \neg x_2 \vee x_3$. A *conjunctive normal form formula* is a set of clauses joined by the AND operator, e.g., $(x_1 \vee \neg x_2 \vee x_3) \wedge (x_1 \vee \neg x_3 \vee x_4)$. A satisfying assignment is an assignment to the variables such that the Boolean formula is true.

The *3-SAT problem* is the problem of deciding if a conjunctive normal form formula with three literals per clause has a satisfying assignment. We will show that conditional sampling from flow models allows us to solve the 3-SAT problem.

We ignore the issue of representing samples from the conditional distribution with a finite number of bits. However the reduction is still valid if the samples are truncated to a constant number of bits.

### A.2   DESIGN OF THE ADDITIVE COUPLING NETWORK

Given a conjunctive normal form with $m$ clauses, we design a ReLU neural network with 3 hidden layers such that the output is 0 if the input is far from a satisfying assignment, and the output is about a large number $M$ if the input is close to a satisfying assignment.

We will define the following scalar function

$$\delta_\varepsilon(x) = \text{ReLU}\left(\frac{1}{\varepsilon}(x - (1 - \varepsilon))\right) - \text{ReLU}\left(\frac{1}{\varepsilon}(x - (1 - \varepsilon)) - 1\right)$$
$$- \text{ReLU}\left(\frac{1}{\varepsilon}(x - 1)\right) + \text{ReLU}\left(\frac{1}{\varepsilon}(x - 1) - 1\right).$$

This function is 1 if the input is 1, 0 if the input $x$ has $|x - 1| \geq \varepsilon$ and is a linear interpolation on $(1 - \varepsilon, 1 + \varepsilon)$. Note that it can be implemented by a hidden layer of a neural network and a linear transform, which can be absorbed in the following hidden layer. See Figure 7 for a plot of this function.

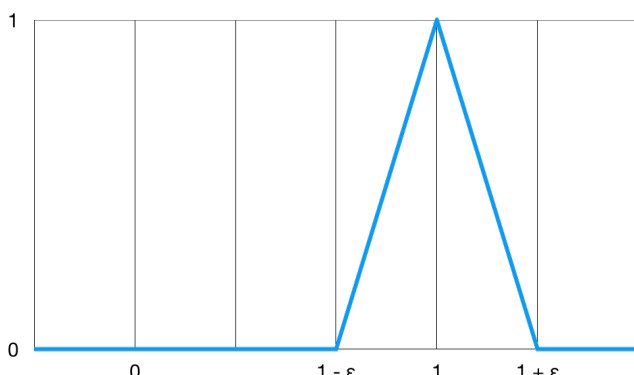

Figure 7: Plot of the scalar function used to construct an additive coupling layer that can generate samples of satisfying 3-SAT assignments.

For each variable $x_i$, we create a transformed variable $\tilde{x}_i$ by applying $\tilde{x}_i = \delta_\varepsilon(x_i) - \delta_\varepsilon(-x_i)$. Note that this function is 0 on $(-\infty, -1 - \varepsilon] \cup [-1 + \varepsilon, 1 - \varepsilon] \cup [1 + \varepsilon, \infty)$, $-1$ at $x_i = -1$, 1 at $x_i = 1$, and a smooth interpolation on the remaining values in the domain.

Every clause has at most 8 satisfying assignments. For each satisfying assignment we will create a neuron with the following process: (1) get the relevant transformed values $\tilde{x}_i, \tilde{x}_j, \tilde{x}_k$, (2) multiply each variable by $1/3$ if it is equal to 1 in the satisfying assignment and $-1/3$ if it is equal to $-1$ in the satisfying assignment, (3) sum the scaled variables, (4) apply the $\delta_\varepsilon$ function to the sum.

We will then sum all the neurons corresponding to a satisfying assignment for clause $C_j$ to get the value $c_j$. The final output is the value $M \times \text{ReLU}(\sum_j c_j - (m - 1))$, where $M$ is a large scalar.

We say that an input to the neural network $x$ corresponds to a Boolean assignment $x' \in \{-1, 1\}^d$ if for every $x_i$ we have $|x_i - x_i'| < \varepsilon$. For $\varepsilon < 1/3$, if the input does not correspond to a satisfying assignment of the given formula, then at least one of the values $c_j$ is 0. The remaining values of $c_j$ are at most 1, so the sum in the output is at most $(m - 1)$, thus the sum is at most zero, so the final output is 0. However, if the input is a satisfying assignment, then every value of $c_j = 1$, so the output is $M$.

### A.3 GENERATING SAT SOLUTIONS FROM THE CONDITIONAL DISTRIBUTION

Our flow model will take in Gaussian noise $x_1, \ldots, x_d, z \sim N(0, 1)$. The values $x_1, \ldots, x_d$ will be passed through to the output. The output variable $y$ will be $z + f_M(x_1, \ldots, x_d)$, where $f_M$ is the neural network described in the previous section, and $M$ is the parameter in the output to be decided later.

Let $A$ be all the valid satisfying assignments to the given formula. For each assignment $a$, we will define $X_a$ to be the region $X_a = \{x \in \mathbb{R}^d : \|a - x\|_\infty \leq \varepsilon\}$, where as above $\varepsilon$ is some constant less than $1/3$. Let $X_A = \bigcup_{a \in A} X_a$.

Given an element $x \in X_a$, we can recreate the corresponding satisfying assignment $a$. Thus if we have an element of $X_A$, we can certify that there is a satisfying assignment. We will show that the distribution conditioned on $y = M$ can generate satisfying assignments with high probability.

We have that
$$p(X_A \mid y = M) = \frac{p(y = M, X_A)}{p(y = M, X_A) + p(y = M, \overline{X}_A)}$$
If we can show that $p(y = M, \overline{X}_A) \ll p(y = M, X_A)$, then we have that the generated samples are with high probability satisfying assignments.

Note that,
$$p(y = M, \overline{X}_A) = p(y = M \mid \overline{X}_A) P(\overline{X}_A) \leq p(y = M \mid \overline{X}_A).$$
Also notice that if $x \in \overline{X}_A$, then $f_M(x) = 0$. Thus $y \sim \mathcal{N}(0, 1)$ and $P(y = M \mid \overline{X}_A) = \Theta(\exp(-M^2/2))$.

Now consider any satisfying assignment $x_a$. Let $X_a'$ be the region $X_a' = \{x \in \mathbb{R}^d : \|a - x\|_\infty \leq \frac{1}{2m}\}$. Note that for every $x$ in this region we have $f_M(x) \geq M/2$. Additionally, we have that $P(X_a') = \Theta(m)^{-d}$. Thus for any $x \in X_a'$, we have $p(Y = M \mid x) \gtrsim \exp(-M^2/8)$. We can conclude that
$$p(y = M, X_A) \geq p(Y = M, X_a') = \int_{X_a'} p(Y = M \mid x) p(x) \, dx \gtrsim \exp(-M^2/8 - \Theta(d \log m)).$$
For $M = O(\sqrt{d \log m})$, we have that $p(y = M, \overline{X}_A)$ is exponentially smaller than $p(y = M, X_A)$. This implies that sampling from the distribution conditioned on $y = M$ will return a satisfying assignment with high probability.

### A.4 HARDNESS OF APPROXIMATE SAMPLING

**Definition 2.** The complexity class $RP$ is the class of decision problems with efficient random algorithms that (1) output YES with probability $1/2$ if the true answer is YES and (2) output NO with probability 1 if the true answer is NO. It is widely believed that $RP$ is a strict subset of $NP$.

A simple extension of the above theorem shows that even approximately matching the true conditional distribution in terms of TV distance is computationally hard. The total variation (TV) distance is defined as $d_{\mathrm{TV}}(p, q) = \sup_E |p(E) - q(E)| \leq 1$, where $E$ is an event. The below corollary shows that it is hard to conditionally sample from a distribution that is even slightly bounded away from 1.

**Corollary 3.** *The conditional sampling problem remains hard even if we only require the algorithm to sample from a distribution $q$ such that $d_{\mathrm{TV}}(p(\cdot \mid x = x^*), q) \leq 1 - 1/\mathrm{poly}(d)$, where $d$ is the dimension of the distribution.*

We show that the problem is still hard even if we require the algorithm to sample from a distribution $q$ such that $d_{\mathrm{TV}}(p(x \mid y = y^*), q) \geq 1/\mathrm{poly}(d)$.

Consider the event $X_A$ from above. We saw that $p(X_A \mid y = M) \geq 1 - \exp(-\Omega(d))$. We have that $d_{\mathrm{TV}}(p(\cdot \mid y = M), q) \geq 1 - \exp(-\Omega(d) - q(X_A))$.

Suppose that the distribution $q$ has $q(X_A) \geq 1/\text{poly}(d)$. Then by sampling a polynomial number of times from $q$ we sample an element of $X_A$, which allows us to find a satisfying assignment. Thus if we can efficiently create such a distribution, we would be able to efficiently solve SAT and RP = NP. As we are assuming this is false, we must have $q(X_A) \leq 1/\text{poly}(d)$, which implies $d_{\text{TV}}(p(\cdot \mid y = M), q) \geq 1 - 1/\text{poly}(d)$.

# B  MISSING DERIVATIONS

## B.1  DERIVATION OF EQUATION (3)

Here we present a detailed derivation of Equation (3). Note that this equality is true *up to a constant w.r.t.* $\hat{f}$, which is fine as we use this as the optimization objective.

$$
\begin{aligned}
\mathcal{L}_{\text{Ours}}(\hat{f}) &\triangleq D_{\text{KL}}(p_{f \circ \hat{f}}(\boldsymbol{x}) \parallel p_f(\boldsymbol{x} \mid \tilde{\boldsymbol{y}} = \boldsymbol{y}^*)) \\
&= \mathbb{E}_{\boldsymbol{x} \sim p_{f \circ \hat{f}}} \left[ \log p_{f \circ \hat{f}}(\boldsymbol{x}) - \log p_f(\boldsymbol{x}, \tilde{\boldsymbol{y}} = \boldsymbol{y}^*) \right] + \log p_f(\tilde{\boldsymbol{y}} = \boldsymbol{y}^*) \\
&\overset{A}{=} \mathbb{E}_{\boldsymbol{x} \sim p_{f \circ \hat{f}}} \left[ \log p_{f \circ \hat{f}}(\boldsymbol{x}) - \log p_f(\boldsymbol{x}) - \log p_\sigma(\tilde{\boldsymbol{y}} = \boldsymbol{y}^* \mid \boldsymbol{x}) \right] \\
&\overset{B}{=} \mathbb{E}_{\boldsymbol{x} \sim p_{f \circ \hat{f}}} \left[ \log p_{f \circ \hat{f}}(\boldsymbol{x}) - \log p_f(\boldsymbol{x}) \right] + \mathbb{E}_{\boldsymbol{x} \sim p_{f \circ \hat{f}}} \left[ - \log p_\sigma(\tilde{\boldsymbol{y}} = \boldsymbol{y}^* \mid \boldsymbol{y} = T(\boldsymbol{x})) \right] \\
&= D_{\text{KL}}(p_{f \circ \hat{f}}(\boldsymbol{x}) \parallel p_f(\boldsymbol{x})) + \mathbb{E}_{\boldsymbol{x} \sim p_{f \circ \hat{f}}} \left[ - \log p_\sigma(\tilde{\boldsymbol{y}} = \boldsymbol{y}^* \mid \boldsymbol{y} = T(\boldsymbol{x})) \right] \\
&\overset{C}{=} D_{\text{KL}}(p_{\hat{f}}(\boldsymbol{z}) \parallel p_f(\boldsymbol{z})) + \mathbb{E}_{\boldsymbol{z} \sim p_{\hat{f}}} \left[ \frac{1}{2\sigma^2} \| T(f(\boldsymbol{z})) - \boldsymbol{y}^* \|_2^2 \right]
\end{aligned}
$$

In $(A)$, we drop the $\log p_f(\tilde{\boldsymbol{y}} = \boldsymbol{y}^*)$ term as it is constant w.r.t. $\hat{f}$.
In $(B)$, we use the conditional independence $\tilde{\boldsymbol{y}} \perp\!\!\!\perp \boldsymbol{x} \mid \boldsymbol{y}$.
In $(C)$, we use the invariance of KL divergence under invertible transformation to rewrite the KL divergence in terms of $\boldsymbol{z}$.

## B.2  JOINT VI VS. MARGINAL VI

We also provide a justification for using the joint VI loss as discussed in section 4. Specifically, we show that the joint VI loss is an upper bound to the intractable marginal VI loss in eq. (1), which we wish to minimize. For notational brevity, we write $q(\boldsymbol{x})$ and $q(\boldsymbol{z})$ to denote the variational posterior $p_{f \circ \hat{f}}(\boldsymbol{x})$ and the pre-generator $p_{\hat{f}}(\boldsymbol{z})$, respectively. Then,

$$
\begin{aligned}
&\text{(Joint KL)} \\
&= D_{\text{KL}}(q(\boldsymbol{x}) \parallel p_f(\boldsymbol{x}|\tilde{\boldsymbol{x}}_1 = \boldsymbol{x}_1^*)) \\
&= \mathbb{E}_{q(\boldsymbol{x}_1, \boldsymbol{x}_2)} [\log q(\boldsymbol{x}_1, \boldsymbol{x}_2) - \log p_f(\boldsymbol{x}_1, \boldsymbol{x}_2 | \tilde{\boldsymbol{x}}_1 = \boldsymbol{x}_1^*)] \\
&= \mathbb{E}_{q(\boldsymbol{x}_1, \boldsymbol{x}_2)} [\log q(\boldsymbol{x}_2) + \log q(\boldsymbol{x}_1 \mid \boldsymbol{x}_2) - \log p_f(\boldsymbol{x}_2 | \tilde{\boldsymbol{x}}_1 = \boldsymbol{x}_1^*) - \log p_f(\boldsymbol{x}_1 \mid \tilde{\boldsymbol{x}}_1 = \boldsymbol{x}_1^*, \boldsymbol{x}_2)] \\
&= \mathbb{E}_{q(\boldsymbol{x}_2)} [\log q(\boldsymbol{x}_2) - \log p_f(\boldsymbol{x}_2 | \tilde{\boldsymbol{x}}_1 = \boldsymbol{x}_1^*)] \\
&\quad + \mathbb{E}_{q(\boldsymbol{x}_2)} \left[ \mathbb{E}_{q(\boldsymbol{x}_1 | \boldsymbol{x}_2)} [\log q(\boldsymbol{x}_1 \mid \boldsymbol{x}_2) - \log p_f(\boldsymbol{x}_1 \mid \tilde{\boldsymbol{x}}_1 = \boldsymbol{x}_1^*, \boldsymbol{x}_2)] \right] \\
&= D_{\text{KL}}(q(\boldsymbol{x}_2) \parallel p_f(\boldsymbol{x}_2 | \tilde{\boldsymbol{x}}_1 = \boldsymbol{x}_1^*)) + \mathbb{E}_{q(\boldsymbol{x}_2)} [D_{\text{KL}}(q(\boldsymbol{x}_1 \mid \boldsymbol{x}_2) \parallel p_f(\boldsymbol{x}_1 | \tilde{\boldsymbol{x}}_1 = \boldsymbol{x}_1^*, \boldsymbol{x}_2))] \\
&\geq D_{\text{KL}}(q(\boldsymbol{x}_2) \parallel p_f(\boldsymbol{x}_2 | \tilde{\boldsymbol{x}}_1 = \boldsymbol{x}_1^*)) \\
&= \text{(Marginal KL)},
\end{aligned}
$$

where the last inequality is due to the nonnegativity of KL. Note that equality holds when $D_{\text{KL}}(q(\boldsymbol{x}_1 \mid \boldsymbol{x}_2) \parallel p_f(\boldsymbol{x}_1|\tilde{\boldsymbol{x}}_1 = \boldsymbol{x}_1^*, \boldsymbol{x}_2)) = 0$, i.e. when our variational posterior matches the true conditional.

# C  EXPERIMENT DETAILS

## C.1  OUR ALGORITHM

## C.2  EFFECTS OF THE SMOOTHING PARAMETER

The choice of variance for Gaussian smoothing is an important hyperparameter, so we provide some empirical analysis of the effects of $\sigma$. As shown in Figure 8a, large values of $\sigma$ cause the samples to ignore the observation, whereas small values lead to unnatural samples as the learned distribution becomes more degenerate. Visually, we achieve essentially negligible variance on the observed

---

**Algorithm 1 Training the pre-generator for a given observation under transformation**. We assume that $\hat{f}$ is an invertible neural network with parameters $\theta$.

---

1: **Input**: $\boldsymbol{y}^*$: observation we are conditioning on, $T(\boldsymbol{x})$: differentiable transformation of $\boldsymbol{x}$.
2: **for** $i = 1 \ldots \texttt{num\_steps}$ **do**
3:      **for** $j = 1 \ldots m$ **do**                      $\triangleright$ generate $m$ latent codes from $p_{\hat{f}}(\boldsymbol{z})$
4:          Sample $\boldsymbol{\epsilon}^{(j)} \sim \mathcal{N}(\boldsymbol{0}, I_d)$
5:          $\boldsymbol{z}^{(j)} \leftarrow \hat{f}(\boldsymbol{\epsilon}^{(j)})$
6:      **end for**
7:      $\mathcal{L} \leftarrow \frac{1}{m} \sum_{j=1}^{m} \left[ \log p_{\hat{f}}(\boldsymbol{z}^{(j)}) - \log p_{\text{normal}}(\boldsymbol{z}^{(j)}) + \frac{1}{2\sigma^2} \left\| T(f(\boldsymbol{z}^{(j)})) - \boldsymbol{y}^* \right\|_2^2 \right]$
8:      $\theta \leftarrow \theta - \nabla_\theta \mathcal{L}$                                       $\triangleright$ gradient step
9: **end for**

---

**Algorithm 2 Sampling from the approximate posterior from the learned pre-generator.**

---

1: **Input**: $f$: base model, $\hat{f}$: pre-generator trained on the given observation, $n$: number of samples.
2: **for** $i = 1 \ldots n$ **do**
3:      Sample $\boldsymbol{\epsilon}^{(i)} \sim \mathcal{N}(\boldsymbol{0}, I_d)$
4:      $\boldsymbol{x}^{(i)} = (\boldsymbol{x}_1^{(i)}, \boldsymbol{x}_2^{(i)}) \leftarrow f(\hat{f}(\boldsymbol{\epsilon}^{(i)}))$          $\triangleright$ Feed the noise through composed flow.
5: **end for**
6: **if** Partitioned Case **then**
7:      Return $\{\boldsymbol{x}_2^{(1)}, \ldots, \boldsymbol{x}_2^{(n)}\}$          $\triangleright$ Return only the unobserved portion of the samples.
8: **else**
9:      Return $\{\boldsymbol{x}^{(1)}, \ldots, \boldsymbol{x}^{(n)}\}$
10: **end if**

---

portion past $\sigma = 0.01$, but at the slight degradation in the sample quality. In Figure 8b, we also notice that the difference between the true observation ($\boldsymbol{x}_1^*$) and generated observation ($\tilde{\boldsymbol{x}}_1$) stops improving past $\sigma = 1e-4$. We tried annealing $\sigma$ from a large value to a small positive target value to see if that would help improve the sample quality at very small values of $\sigma$, but noticed no appreciable difference. In practice, we recommend using the largest possible $\sigma$ that produces observations that are within the (task-specific) acceptable range of the true observation.

### C.3    HYPERPARAMETERS: BASE MODEL AND PRE-GENERATOR

See Table 3 and Table 4 for the hyperparameters used to define the network architectures train them. For the color datasets CIFAR-10 and CelebA-HQ, we used 5-bit pixel quantization following Kingma & Dhariwal (2018). Additionally for CelebA-HQ, we used the same train-test split (27,000/3,000) of Kingma & Dhariwal (2018) and resized the images to $64 \times 64$ resolution. Uncurated samples from the base models are included for reference in Figure 9.

Table 3: Hyperparameters used to train the base models used in our experiments.

| Base Models | MNIST | CIFAR-10 | CelebA-HQ |
|---|---|---|---|
| Image resolution | $28 \times 28$ | $32 \times 32$ | $64 \times 64$ |
| Num. scales | 3 | 6 | 6 |
| Res. blocks per scale | 8 | 12 | 10 |
| Res. block channels | 32 | 64 | 80 |
| Bits per pixel | 8 | 5 | 5 |
| Batch size | 128 | 64 | 32 |
| Learning rate | 0.001 | 0.001 | 0.001 |
| Num. epochs | 200 | | |
| Test set bits-per-dim | 1.053 | 1.725 | 1.268 |

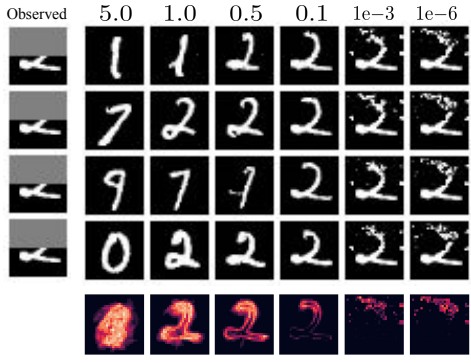

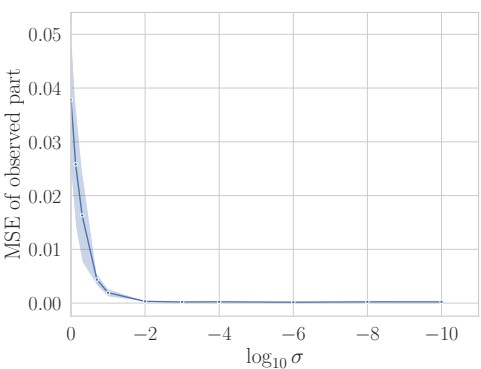

(a) Each column contains samples from the learned conditional sampler at different values of $\sigma$ with pixelwise variance computed using 32 samples.

(b) MSE between $\tilde{x}_1$ and $x_1^*$ at different values of $\sigma$.

Figure 8: Effect of the smoothing parameter on sample quality and tightness of approximate conditioning.

Table 4: Hyperparameters used to define and train the pre-generator for each of our experiments.

| Base Models | MNIST | CIFAR-10 | CelebA-HQ |
|---|---|---|---|
| Image resolution | $28 \times 28$ | $32 \times 32$ | $64 \times 64$ |
| Num. scales | 3 | 4 | 3 |
| Res. blocks per scale | 3 | 4 | 3 |
| Res. block channels | 32 | 48 | 48 |
| Batch size | 64 | 32 | 8 |

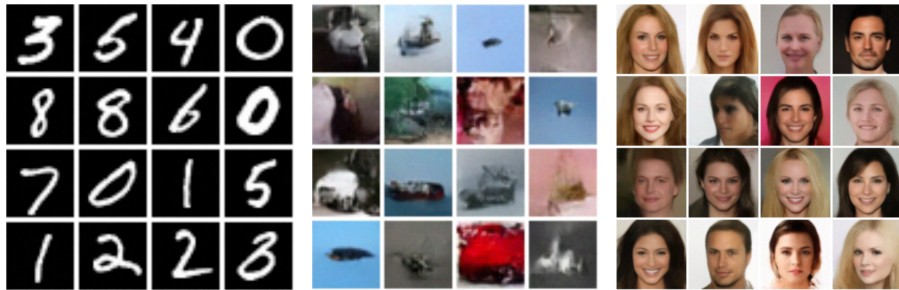

Figure 9: Unconditional samples from the base models used for our experiments. From left: MNIST, 5-bit CIFAR-10, and 5-bit CelebA-HQ models.

### C.4 HYPERPARAMETERS: IMAGE INPAINTING

We randomly chose 900/500/300 images from MNIST/CIFAR-10/CelebA-HQ test sets, applied masks defined in Section 6.1, and generated samples conditioned on the remaining parts. FID and other sample quality metrics were computed using 32 conditional samples per test image for VI methods (ours and Ambient VI), 8 for Langevin Dynamics, and 6 for PL-MCMC. We note that using more samples for VI methods do not unfairly affect the result of sample quality evaluation, i.e. there was no appreciable difference when using 8 vs. 32 samples to compute FID. We used more samples simply because it is much cheaper for VI methods to generate samples compared to MCMC methods.

**For VI Methods (Ours & Ambient VI)**

- Learning rate: $1e-3$ for MNIST; $5e-4$ for the others
- Number of training steps: 2000 for CelebA-HQ; 1000 for the others

**For Langevin Dynamics**

- Learning rate: 5e−4 for all datasets
- Length of chain: 1000 for CIFAR-10; 2000 for the others

**For PL-MCMC**

- Learning rate: 5e−4
- Length of chain: 2000 for MNIST
- $\sigma_a = 1e-3$, $\sigma_p = 0.05$

## C.5 HYPERPARAMETERS: INVERSE PROBLEMS

|  | Colorization | Compressed Sensing | Compressed Sensing | Super-resolution |
|---|---|---|---|---|
| Learning rate | 5e−4 | 5e−4 | 5e−4 | 5e−4 |
| $\sigma$ | 0.05 | 0.05 | 0.05 | 0.05 |
| Dataset | CelebA-HQ | CelebA-HQ | CIFAR-10 | CIFAR-10 |
| Batch size | 8 | 8 | 32 | 32 |
| Number of steps | 1000 | 2000 | 1000 | 1000 |

Engel et al. (2017); Parno & Marzouk (2018); Hoffman et al. (2019); Nijkamp et al. (2020)

