# OpenReview forum: "Approximate Probabilistic Inference with Composed Flows"
_ICLR.cc/2021/Conference — Reject_

### Official Review · AnonReviewer4 · 2020-10-26
**A standard use of stochastic variational inference with a not motivated use of likelihood smoothing**

**Rating:** 4
**Confidence:** 4

**Review:**

Summary:
This paper is concerned with the approximation of conditional densities in trained normalizing flow models. The author use a variational Bayesian approach to estimate the conditional density of a set of variables given another set. As use cases, the authors present results in image inpainting and colorization. The paper also contains a theoretical result showing that exact conditioning is NP hard in additive normalizing flow models.

Pros:
- Relevant problem.
- Possibly interesting theoretical result.
- Interesting range of experiments with careful analysis of both quantitative and qualitative results. Large and appropriate list of performance metrics

Cons:
- Low originality, the main idea to perform inference in the latent space is obvious
- The relevance of the theoretical result in justifying the rest of the paper is not clear and not enough space is given to its explanation.
- The smoothing of the likelihood using a very rough noise model is not required
- More baselines are needed. In particular, it would be useful to have baselines without likelihood smoothing and also comparison with (simulation based) forward KL methods such as [1,3].

Relevance:
The problem of conditioning in generative models is highly relevant in our current ML environment as it allows to convert generators into very flexible inference machines capable of solving a large variety of problems.

Originality: With the possible exception of the NP-hardness theorem, the original contributions of this work are very limited with some questionable elements (see below).  Conditioning deep differentiable models is a standard domain of application of variational Bayesian inference and the authors use a very standard and natural  approach. The main idea behind the method is to perform inference in the latent space, this in my opinion is not a noteworthy contribution as it is just the obvious way to do inference in this setting. The second methodological trick is to use a Gaussian smoothing of the likelihood. To my understanding, this procedure is neither well motivated nor supported experimentally.

Major concerns:
I could be missing something but I do not understand why the authors are smoothing the likelihood since the flow already give a perfectly well-behaved joint model. The authors also seem to motivate this choice using their theorem showing that exact conditioning is NP hard. However this is only a valid motivation for using approximate inference, not for adopting an approximate likelihood. It seems that the authors want to get a least-square loss component in the pixel (or add-hoc feature) space. Without smoothing you will get a least square loss in the latent space which is likely to be much more appropriate. Therefore, I am not convinced that this work better in practice and it should at least be tested experimentally.

Paper structure:
In general, the paper is well structured with a clear narrative and an appropriate amount of background material.
However, the treatment of the theorem in section 3 should be expanded. The experiment section is very well structured and the analysis of the results is very good and definitely above average. I am pleased that the authors took the time to analyze the bad performance of the native VI method.

Writing:
The paper is very clearly written.

Literature
The coverage of the literature is appropriate. However, the author should also discuss methods based on synthetic sampling and forward KL divergence as they are a very viable approach to generative model conditioning in the latent space. For example:

[1] Papamakarios, George, and Iain Murray. "Fast ε-free inference of simulation models with bayesian conditional density estimation." Advances in Neural Information Processing Systems. 2016.
[2] Le, Tuan Anh, Atilim Gunes Baydin, and Frank Wood. "Inference compilation and universal probabilistic programming." Artificial Intelligence and Statistics. PMLR, 2017.
[3] Ambrogioni, Luca, et al. "Forward amortized inference for likelihood-free variational marginalization." The 22nd International Conference on Artificial Intelligence and Statistics. PMLR, 2019.

---

> ### Author Response · Authors · 2020-11-19
> **Response to Reviewer 4 (Part 1)**
>
> We thank R4 for the detailed and constructive feedback with pointers to additional references.  Below we include our response to the concerns raised by R4.
>
> **1. Clarification of the main contribution of the paper.**
> As mirrored in our response to R1, we clarify that the main contribution of our work is not the idea of performing inference in the latent space.  Rather, our key idea is the use of a pre-generator **parametrized as a flow**, which can then be trained via variational inference by exploiting the invertibility of the base model. We hope that Section 5 (Related Works) in the updated manuscript makes this point clearer, and also refer to our first response to R1 for additional context.
>
> **2. Implications of the hardness proof and the necessity for smoothing.**
> As R3 mentioned, we believe there may be a misunderstanding around the implications of our hardness result. We clarify a few points below.
>
> > Hardness of exact conditioning is only a valid motivation for using approximate inference, not for adopting an approximate likelihood.
>
> While this statement is true if we only assume Theorem 1, our hardness result goes a bit further. As stated in Section 3 and in Corollary 3 (Appendix A), even **approximating** (w.r.t total variation distance) the true conditional distribution $p(x_2 \mid x_1=x^*)$ is hard as long as we require exact conditioning. This is what motivates our relaxation to allow approximate conditioning via smoothing, i.e. we aim to learn $p(x_2  \mid x_1 \approx x^*)$ (note the change of $x_1=x^*$ to $x_1 \approx x^*$). We have updated Section 3 to further emphasize this point.
>
> > It seems that the authors want to get a least-square loss component in the pixel space.
>
> Given the above motivation for approximate conditioning, the particular choice of smoothing scheme is a hyperparameter that needs to be chosen.  We have experimented with Gaussian smoothing as well as non-Gaussian smoothing, such as the energy-based kernel implicitly defined by the LPIPS distance $p(\tilde{x}_1 \mid x_1) \propto \exp(-\text{LPIPS}(\tilde{x}_1, x_1))$.  The intuition was that we could achieve better sample quality by relying on a perceptual metric.  However there was no appreciable difference between the two in our preliminary experiments, so we chose to use Gaussian smoothing simply because it was faster to run experiments with. Thus, the least square term in the loss was a result of using Gaussian smoothing, not a choice that motivated the use of Gaussian smoothing. It is possible that better performance can be achieved with more extensive search over the smoothing distribution, but we leave that investigation for future work.
>
> > Without smoothing you get a least square loss in the latent space which is likely to be much more appropriate. Gaussian smoothing is neither well motivated nor supported experimentally.
>
> We appreciate your suggestion, but it's unclear to us how not smoothing the observation would lead to a least square loss in the latent space.  In the general case of conditioning under the transformation $y=T(x)$, the latent VI loss without smoothing simplifies to $D_{\rm KL} (p_{\hat{f}}(z) \Vert p_f(z \mid y=y^*)) = D_{\rm KL}(p_{\hat{f}}(z) \Vert p_f(z)) - \mathbb{E}_{z \sim q}[\log p(y=y^* \mid z)] + \log p(y=y^*)$, which to the best of our knowledge does not contain a least square term.  Moreover, the second term is particularly problematic for optimization because $T \circ f$ is a deterministic function and $\log p(y=y^* \mid z)$ is degenerate, i.e. it is undefined for all $z$ such that $T(f(z))$ does not exactly match the observation $y^*$.  We believe this, combined with the hardness result, sufficiently motivates the use of smoothing.
> If there is any misunderstanding on our part on this point, we'd greatly appreciate a clarification.

---

> > ### Author Response · Authors · 2020-11-19
> > **Response to Reviewer 4 (Part 2)**
> >
> > **3. More baselines are needed. In particular, it would be useful to have baselines without likelihood smoothing and forward KL methods such as [1,3].**
> > We thank R4 for suggesting additional references [1-3] that propose interesting solutions to difficult inference tasks.  That said, we believe forward KL methods are not applicable in our setting, which we explain in the context of [3].  In [3], the authors get around the intractability of forward KL variational inference by performing joint-contrastive VI, from which the forward amortized VI (FAVI) loss (equation 6 in [3]) is derived. Written using notations from our setup, this loss is $L_{FA} = \mathbb{E}_{p(x_1,x_2)}[-\log q(x_2 \mid x_1)]$.
> >
> > There are two important distinctions from our approach.  First, this loss is only valid for the amortized version of our setup where we wish to train a single variational posterior $q$ for all observations $x_1^*$.  Indeed Section 4.1 of [3] mentions that FAVI can be derived as an amortization of the stochastic forward VI loss $D_{\rm KL} (p(x_2 \mid x_1) \Vert q(x_2 \mid x_1))$. This is different from our approach based on stochastic VI, which doesn't suffer from amortization gap [4].  Second, the FAVI formulation does not leverage the invertibility of the base model. In the extreme case where we condition on a degenerate, constant observation $T(x) = c$, the minimizer of FAVI loss is $p(x)$ itself: $p(x) = \arg\max_q D_{\rm KL} (p(x \mid c) \Vert q(x \mid c))$. Thus FAVI attempts to distill the base model using the variational posterior, which can be a challenging optimization task. For our method, this corresponds to the trivial task of $\hat{f}$ learning to represent the identity function. While it may be possible to adapt [3] to leverage the base model, we believe this is outside the scope of our work.
> >
> > **References**
> > [1] Papamakarios, George, and Iain Murray. "Fast ε-free inference of simulation models with bayesian conditional density estimation." Advances in Neural Information Processing Systems. 2016.
> > [2] Le, Tuan Anh, Atilim Gunes Baydin, and Frank Wood. "Inference compilation and universal probabilistic programming." Artificial Intelligence and Statistics. PMLR, 2017.
> > [3] Ambrogioni, Luca, et al. "Forward amortized inference for likelihood-free variational marginalization." The 22nd International Conference on Artificial Intelligence and Statistics. PMLR, 2019.
> > [4] Cremer, C., Li, X. & Duvenaud, D.. (2018). Inference Suboptimality in Variational Autoencoders. Proceedings of the 35th International Conference on Machine Learning, in PMLR 80:1078-1086

---

### Official Review · AnonReviewer3 · 2020-10-26
**A solid contribution to the field of normalizing flows and conditional generative models**

**Rating:** 7
**Confidence:** 4

**Review:**

In this work, the authors propose a novel method of estimating conditional distributions over arbitrary partitions of variables $x = [x_1,x_2]$ using an existing pre-trained flow model for $p(x)$. Their method fits a new *pre-generator* flow $\hat{f}$ for each observation which maps from a base distribution $\epsilon \sim N(0,\mathbb{I})$ to the latent variables $z \sim N(0,\mathbb{I})$ where the mapping $z \leftrightarrow x$ is learned by the pre-trained "base" flow $f$. The result is a mapping $\hat{f}$ which learns to shift probability mass to regions of the latent space which correspond to the conditional distribution of the given observation. The authors present comprehensive experimental results which show a clear improvement over existing methods for conditional inference but with the drawback of needing to re-train the pre-generator for each individual observation.

Pros:
- Very well written, clear, easy to understand
- The proposed method is intuitive and well defined
- Placement of the method relative to recent work is very clearly explained
- Comprehensive theoretical analysis including an interesting hardness result, which is uncommon for the deep learning literature
- Comprehensive and convincing empirical analysis with clear results

Cons:
- There is very little discussion on how the construction of the base generator $f$ affects the results of the proposed method
- The proof of hardness is somewhat opaque and feels contrived; but this is often the case with hardness proofs!
- The method has a clear weakness in needing to be retrained for each observation. However, this is clearly stated by the authors and left open as a direction for future work.

Overall, I think this is an exceptional paper which makes a significant contribution to the field. I think it is suitable to accept as-is with only a few minor adjustments which I will enumerate below.

1. It would be nice (but not absolutely necessary) to see some discussion regarding the construction of the base generator, as I mentioned in the Cons above; e.g. does the performance of this method depend significantly on the user's choice of base model? Intuitively, I would think so.
2. A few notes on the proofs:
- The variable lower-case $m$ shows up in several spots in the hardness proof but is never defined. Perhaps these are typos and you meant to write $M$?
- In the proof for equation 3, the notation for expectations (i.e. $\mathbb{E}$) is inconsistent in a few places. Presumably just typos.
- I may be missing something, but it's not immediately clear why $y=T(x)$ can be substituted for the conditioner $x$ in $p_{\sigma}(\tilde{y}=y^*|x)$. My immediate intuition is that this would only be valid if $T$ is injective, otherwise this may change the underlying conditional density. Please correct me if I am wrong, and preferably add a clarification to the proof as to why this is justified.

Congratulations to the authors on a job well done!

---

> ### Comment · AnonReviewer3 · 2020-11-12
> **Updated score after checking reviewer references**
>
> After reviewing the citations provided by reviewer 1 (in particular reference [1] which I had not previously seen), I must agree that the novelty of the authors' approach is more questionable than I had originally thought. Thus, the proposed use of a flow as a "pre-generator" is more incremental than innovative.
>
> I do not think that references 2, 3, and 4 are as relevant as reviewer 1 suggests.
>
> I do not agree with objections from reviewers 1 and 4 that the smoothing is unnecessary. The relaxation of the conditional inference problem follows naturally from the authors' theoretical result proving the hardness of conditional inference under arbitrary partitions of variables. This makes intuitive sense as well since a fixed (non-smooth) observation will induce a degenerate conditional distribution. Reviewer 1 points out that it might not be relevant in the case when the observation is a non-deterministic function of x. This appears to me to be outside of the scope of this work and an irrelevant objection.
>
> In light of this, I am updating my review score to a 7. I still believe this is a good paper and should be seriously considered for acceptance. This is my final score, and I will defend it as necessary.

---

> > ### Comment · AnonReviewer1 · 2020-11-18
> > **Justification of the score and explanation of relevancy**
> >
> > Thank you for taking your time to read the other reviews including mine.
> >
> > Ref [2-3] are prior work using a flow to improve the geometry of the energy for performing gradient-based MCMC. I deem them relevant because this work proposes to compose a latent flow with a trained flow generator to perform inference, which is equivalent to performing MCMC in the latent space. The fact that the latent flow can outperform the ambient VI or ambient Langevin MCMC is not surprising, as this problem has been demonstrated and mitigated in the cited work. This calls the novelty of this paper into question (contribution (A) in my label), which is why I requested stronger baselines in the experiment section; and it also supports the following quote from R4:
> >
> > > The main idea behind the method is to perform inference in the latent space, this in my opinion is not a noteworthy contribution as it is just the obvious way to do inference in this setting.
> >
> > One could argue performing latent VI is not exactly the same as performing latent space MCMC. But then a better motivation of why VI is needed should be provided, e.g. the option to perform amortization to speed up inference (which is not conducted in this work).
> >
> > **Smoothing**: I did not say smoothing is unnecessary. I understand without the smoothing the hard constraints will result in degenerate target distributions. I only mentioned the softened constraints might not be desirable and asked if it's possible to anneal it away (for example, so that the border of the inpainted images will look closer to that of the original images).

---

> > ### Author Response · Authors · 2020-11-19
> > **Response to Reviewer 3**
> >
> > We thank R3 for the thoughtful and positive feedback. We are glad that you found our work valuable, and have incorporated the suggestions in the updated manuscript. More detailed response to the questions raised by R3 are included below.
> >
> > **1. Effect of the base model on the performance of the overall method.**
> > We are glad that R3 pointed this out, as we also think that this is an important aspect we didn't get to investigate further.  While our setup assumes that the base model $p_f(x)$ is correct, in practice there will always be a mismatch between $p_f(x)$ and $p_{\rm true}(x)$.  Interestingly, all our experiments were done using images from the test set, i.e. samples from $p_{\rm true}(x)$, not the base model $p_f(x)$.  The fact that resulting conditional samples look reasonable shows some level of robustness of our method to model mismatch.  We'd like to probe deeper into this by repeating our experiments using multiple base models of varying quality (e.g. measured by test set bpd), but we leave it as future work for now.
> >
> > **2. It's not immediately clear why $y=T(x)$ can be substituted for the conditioner $x$ in $p_{\sigma} (\tilde{y} = y^\* \mid x)$.**
> > This is simply the result of rewriting the conditioning on $x$ in terms of the observed variable $y \triangleq T(x)$.  To see this, notice that by definition the density $p(y \mid x)$ is the Dirac delta function $\delta_{T(x)}(y)$ concentrated at $T(x)$. Thus $p_\sigma (\tilde{y}=y^* \mid x) = \int_{y} p_\sigma(\tilde{y}=y^* \mid y) p(y \mid x) \mathrm{d}y = \int_{y} p_\sigma(\tilde{y}=y^* \mid y) \delta_{T(x)}(y) \mathrm{d}y = p_\sigma(\tilde{y}=y^* \mid y=T(x))$, where the first equality is true from the conditional independence between $\tilde{y}$ and $x$ given $y$ (as shown in Figure 2c). We clarify this in the updated proof.
> >
> > **3. The variable lower-case $m$ in the hardness proof is never defined.  Also the notation for expectation is inconsistent in the proof of equation 3.**
> > Thank you for catching these. The lowercase $m$ refers to the number of clauses in the given conjunctive normal form, and we define this in the updated manuscript. We also fixed the inconsistent notation for expectation in the proof of equation 3.

---

### Official Review · AnonReviewer1 · 2020-10-27
**Solving interesting inference problems with existing tricks**

**Rating:** 5
**Confidence:** 5

**Review:**

Summary: The paper proposes to solve the conditional inference problem by performing a relaxed version of variational inference in the prior space of the flow-based model. The model p(x) is pretrained, and one is interested sampling from p(x|observation). The observation could be some subset of x (inpainting), gray scale representation of an image (colorization), lower-res representation (super resolution), or noisy version of the data (compressed sensing). The paper proposes to perform inference in the prior space, by composing the post-hoc trained latent flow with the trained invertible decoder, as the conditional distribution in that space is believed to have a better geometry.

Contributions and novelties: (A) propose to perform inference in the latent space of a latent variable model to side step the bad geometry, (B) propose to replace hard constraint with stochastic relaxation (placing an additional likelihood term to model the dummy variable). (C) Applications seem interesting for testing the quality of an unconditional flow-based model.

Flaws: missing several important references in the discussion to prior works. These include [1], which describes a more general framework to post-hoc perform sampling from a conditional distribution of a learned latent variable model by fitting a distribution in the latent space; [2,3,4] which propose to mitigate the bad geometry of the learned data energy (in this case, the density model itself) by transforming it into a space where it’s more Gaussianized. Similar idea has also been incorporated in [5] to enable the training of a residual EBM (no need to cite). The key novelties (A and B) are incremental in nature given the above related works that are not cited.


[1] Latent Constraints: Learning to Generate Conditionally from Unconditional Generative Models, 2017
[2] Transport map accelerated markov chain Monte Carlo, 2014
[3] NeuTra-lizing Bad Geometry in Hamiltonian Monte Carlo Using Neural Transport, 2019
[4] Learning Energy-based Model with Flow-based Backbone by Neural Transport MCMC, 2020
[5] VAEBM: A Symbiosis between Variational Autoencoders and Energy-based Models, 2020

Additional details:

For experiments, please report the bits-per-dim (BPD) of the pre-trained flow model, as well as unconditional samples for reference and better comparison.
“Ambient” langevin dynamics is a very weak baseline, since one can easily improve the mixing via deriving a kernel in the latent space (as per [3]). It wouldl be a more fair comparison since the proposed method also composes with the learned decoder flow. Same goes to the naive implementation of the PL-MCMC which only uses a random walk kernel, which is a very weak baseline.
Are the rows of Fig 4 independent? Why do they have the same stroke style if they are all conditioned only on the label only? Is it an indicator of the mode seeking behavior of the reverse KL?
In the qualitative results (tab 2, fig 3 and fig 5), the generated samples do not really satisfy the hard constraints that they are conditioned on (e.g. the subsets of x for inpainting), possibly due to the relaxation via the dummy variable. This is not desirable. Is it possible to anneal the $\sigma$ while training the latent flow so that it will concentrate on the (potentially degenerate) solution that satisfies these constraints?
The authors claim the hardness result is surprising. It has been long shown that sampling from a general Bayesian Belief Network is NP-hard [6]. Can’t the same conclusion be derived from that, with the main difference being the explicit parameterization via a coupling flow?

[6] The Computational Complexity of Probabilistic Inference Using Bayesian Belief Networks, Cooper, 1990



Additional questions about the hardness result (I didn’t read the proof):

Generality of the hardness result: the statement is about the hardness of sampling from the conditional distribution. What is the conditional distribution in this context, is it referring to p(x2|x1) (i.e. observation is a subset of x)? Please be precise. If this is the case it is consistent with the presentation of the previous section (VI). However the proposed method seems to require a more general treatment to take account of the other tasks, e.g. inverse problems.

The discussion of hardness seems to be used to motivate the relaxation of the hard constraints (of the givens). It doesn’t seem that relevant when the observation is not a deterministic function of x (e.g. inpainting, coloration, etc). For compressed sensing for example, the likelihood p(observation|x) naturally exists and is non-degenerate (by assumption). The presentation seems a bit confusing.


--- POST REBUTTAL ---

Modified my score after the rebuttal, since (1) I believe the re-purposing achieved by this work can potentially broaden the applicability of flow-based generative model (2) the authors have toned down the abstract and clarified the contributions in the intro, which now better reflects the value of the work.

I am still leaning towards rejection at the end given the limited originality of the proposed method and the lack of a more comprehensive discussion of different possible approaches, but as means to the same end. For example, the relaxed inference problem can be solved with an MCMC method. These should all be discussed and compared if the contribution is about repurposing a joint likelihood model using flows.


PS. the last line (the references) of the last page might have been a mistake.

---

> ### Author Response · Authors · 2020-11-19
> **Response to Reviewer 1 (Part 1)**
>
> We thank R1 for the insightful and detailed feedback. We enjoyed going over the additional references and updated our manuscript accordingly.  Below we include our detailed response to the concerns and questions raised by R1.
>
> **1. Incremental novelty given the missing references in discussion of prior works.**
> Again we very much appreciate the references [1-4] relevant to our work.  The revised manuscript now includes a discussion of the said papers in Section 5.
>
> Regarding the concerns for novelty, we emphasize that the key contribution of our work is the use of a **flow-based** pre-generator that is trained with a likelihood-based objective by exploiting the **invertibility** of the base model. Importantly, the resulting conditional sampler retains the computational flexibility of flow models and can be used for tasks (e.g. compression) that require exact conditional likelihood or inversion.  This is clearly different from the setting of [1] where an **adversarially** trained actor network is composed with a **non-invertible** generator and therefore does not offer the same level of computational flexibility. The updated manuscript clearly delineates these distinctions from [1]. We also agree with R3 that the references [2-4] are only marginally relevant to our work. While [2-4] share the general principle of leveraging the favorable geometry of the latent space, their main focus is on improving the mixing of Markov chains and does not benefit from the particularities of our setup, e.g. the invertibility of the base model.
>
> To reiterate: Our method employs VI to learn the variational posterior, which is itself a flow model and provides efficient sampling, inversion, and likelihood estimation -- tasks that are much slower or more difficult with MCMC methods. Thus, our method makes the following trade-off: it gains the computational flexibility of flow models with reasonable empirical performance at the cost of losing asymptotic guarantees of MCMC methods and limiting the variational family to flow models. VI plays a key role here because it allows for a tractable likelihood-based optimization of the pre-generator.
>
> **2. Bits-per-dim and unconditional samples of the pre-trained flow models.**
> We added test set BPD as well as unconditional samples from all three base models used in our experiments in Appendix C.3.
>
> **3. Baselines are weak and can be improved by deriving a kernel in the latent space as per [3].**
> We agree with R1 that the MCMC-based baselines can be further improved.  We again note that the point of our experiments is not to show that our approach can generate better samples, as explicitly mentioned in the introduction: "... VI allows for likelihood evaluation and fast sampling, but at a _lower sample quality_ compared to MCMC counterparts".  Given sufficient mixing time with a well-tuned kernel, we indeed expect MCMC methods to produce better samples than our VI-based approach, whose optimality relies on optimizing a loss function parametrized by a complex neural network. Rather, our experiments provide an empirical evidence that our approach can achieve reasonable sample quality competitive to simple MCMC methods, while retaining other computational benefits of flow.
>
> We also point out that our method makes a fundamentally different trade-off in terms of practical run time when generating i.i.d. samples from the posterior. MCMC methods take time linear in the number of samples per observation due to mixing and autocorrelation, but multiple chains can be easily run in parallel for a batch of observations. Our method takes time linear in the number of observations because each observation requires training a separate pre-generator. But once trained, sampling is essentially instant as each sample only requires a single forward-pass of the composed network $f\circ \hat{f}$, with the benefit of generated samples guaranteed to be i.i.d.
>
> **4. Why do the samples in Fig. 4 have the same stroke style? Is it an indicator of the mode seeking behavior of the reverse KL?**
> This is an interesting observation, and we thank R1 for pointing this out. We believe this was due to two reasons: (1) the MNIST classifiers were not calibrated and overfit to certain stroke styles, and (2) we used a very conservative smoothing parameter of $\sigma = 0.02$.  We ran another version of this experiment with $\sigma = 0.1$ and employed early stopping to avoid overfitting to the classifier output.  The result can be seen in the updated Figure 4a, with samples showing much more diversity.

---

> > ### Author Response · Authors · 2020-11-19
> > **Response to Reviewer 1 (Part 2)**
> >
> > **5. Generated samples not satisfying the hard constraints is not desirable.  Is it possible to anneal the $\sigma$ while training the latent flow so that it will concentrate on the (potentially degenerate) solution that satisfies these constraints?**
> > While it is certainly desirable to generate samples that match the hard constraints perfectly, being able to do so in practice is very unlikely given our hardness result. We have nonetheless experimented with different values of $\sigma$ as well as annealing it to a small positive target variance (since the loss is undefined when $\sigma = 0$).  Empirically, we observed that annealing had no benefit compared to simply training with the target variance from the start.  For very small values of $\sigma$ (e.g. $\sigma < 1\mathrm{e}{-4}$), pixelwise variances for the observed portion were either zero or very low. As shown in Figure 8a, the smoothed observation $\tilde{x}_1$ is visually indistinguishable from $x_1$ for small values of $\sigma$. These findings are summarized in Appendix C.2, with samples at different values of $\sigma$ and plots of pixelwise variance as well as MSE of the observed portions (see Figure 8).
> >
> > **6. Could the hardness result be derived from the result on general Bayesian Belief Network?**
> > The classical paper [8] shows that inference is NP-hard for general directed graphical models, where a probability table is specified for each variable conditioned on its parents.
> > Because these conditional distributions need complexity exponential in the number of parents configurations to describe, directed graphical models limit the in-degree of variables (e.g. to logarithmic in the total number of variables).
> > Thus directed graphical models for which inference is hard necessarily have conditional independence assumptions, while deep generative models do not exhibit such conditional independence unless explicitly enforced. Therefore hardness results for one family of distributions do not transfer to the other.
> >
> > **7. Is the conditional distribution in the hardness resulting referring to $p(x_2 \mid x_1)$?**
> > Yes, Theorem 1 refers to sampling from the condition distribution $p(x_2 \mid x_1)$. We updated the theorem statement to be more precise. Note that our general formulation under differentiable transformation includes this as a special case where $T(x) = x_1$ and sampling procedure simply takes the $x_2$ portion from the joint samples conditioned on the observation.
> >
> > **8. The discussion of hardness seems to be used to motivate the relaxation of the hard constraints (of the givens). It doesn’t seem that relevant when the observation is not a deterministic function of x (e.g. compressed sensing).**
> > Our formulation is indeed specific to deterministic forward operators. We focus on this setting as it already encompasses a large set of tasks, including noiseless compressed sensing, inpainting, super-resolution, phase retrieval, and numerous other inverse problems. It is an established topic in both classical and deep inverse problem literature (e.g. see [5-7]).  Extending our method to noisy/stochastic forward operators would certainly be an interesting research direction, but we agree with R3 that it is out of scope of this work.
> >
> > **References**
> >
> > [1] Jesse  Engel,  Matthew  Hoffman,  and  Adam  Roberts.   Latent  constraints:  Learning  to  generate conditionally from unconditional generative models. arXiv preprint arXiv:1711.05772, 2017.
> > [2] Matthew D Parno and Youssef M Marzouk. Transport map accelerated markov chain monte carlo. SIAM/ASA Journal on Uncertainty Quantification, 6(2):645–682, 2018.
> > [3] Matthew Hoffman, Pavel Sountsov, Joshua V Dillon, Ian Langmore, Dustin Tran, and Srinivas Vasudevan. Neutra-lizing bad geometry in hamiltonian monte carlo using neural transport. arXiv preprint arXiv:1903.03704, 2019.
> > [4] Erik Nijkamp, Ruiqi Gao, Pavel Sountsov, Srinivas Vasudevan, Bo Pang, Song-Chun Zhu, andYing Nian Wu. Learning energy-based model with flow-based backbone by neural transport mcmc. arXiv preprint arXiv:2006.06897, 2020.
> > [5] Ashish Bora, Ajil Jalal, Eric Price, and Alexandros G Dimakis. Compressed sensing using generativemodels. InInternational Conference on Machine Learning, pp. 537–546. JMLR. org, 2017.
> > [6] Lynton Ardizzone, Jakob Kruse, Sebastian J. Wirkert, Daniel Rahner, Eric W. Pellegrini, Ralf S. Klessen,
> > Lena Maier-Hein, Carsten Rother, and Ullrich Köthe. Analyzing inverse problems with invertible neural
> > networks. CoRR, abs/1808.04730, 2018. URL http://arxiv.org/abs/1808.04730.
> > [7] Muhammad Asim, Ali Ahmed, and Paul Hand. Invertible generative models for inverse problems:
> > mitigating representation error and dataset bias. CoRR, abs/1905.11672, 2019. URL http://arxiv.
> > org/abs/1905.11672.
> > [8] Gregory Cooper. The Computational Complexity of Probabilistic Inference Using Bayesian Belief Networks. Artificial Intelligence, Volume 42, Issue 2-3, pp. 393-405, 1990.

---

> > ### Comment · AnonReviewer1 · 2020-11-23
> > **Response to rebuttal**
> >
> > > While [2-4] share the general principle of leveraging the favorable geometry of the latent space, their main focus is on improving the mixing of Markov chains and does not benefit from the particularities of our setup, e.g. the invertibility of the base model.
> >
> > quoted from the abstract:
> >
> > """ We experimentally demonstrate that our approach outperforms Langevin Dynamics in terms of sample quality, while requiring much fewer parameters and training time compared to regular variational inference. """
> >
> > [2-4] also use an invertible map to improve the geometry of the energy, just like how the proposed composed flow uses a pre-trained generator and composes it with a 'pre-generator'. The difference here is that the invertible map is given for free, much like the concurrent work VAEBM. This is the reason why I believe a comparison with LD and vanilla VI in the data space is a bit straw-man. Furthermore, it is reiterated in the response to R4 that "the main contribution of our work is not the idea of performing inference in the latent space"; however, the above quote seems to rely heavily on the benefit of performing inference in the latent space. I found this inconsistent.
> >
> > Also, I agree [1] is fundamentally different from the proposed method, due to its adversarial nature. And I think we all agree it's relevant. But I don't see why "[it] does not offer the same level of computational flexibility".
> >
> > > Thus, our method makes the following trade-off: it gains the computational flexibility of flow models .... VI plays a key role here because it allows for a tractable likelihood-based optimization of the pre-generator.
> >
> > What computational flexibility exactly? Optimizing the proposal distribution via minimizing KL and sampling using Langevin-based MCMC are very similar in practice, as the latter can be seen as a non-parametric flow [*]. This suggests the evolution of the optimization of the proposal for VI & the Markov chain of Langevin-based MCMC are actually very similar to each other.
> >
> > [*] Langevin Dynamics as Nonparametric Variational Inference
> >
> > I do like the general idea of using a pre-trained likelihood-based model and use it to perform various inference tasks. And I think this is relatively under-explored in the literature. But the arguments (of the claimed contributions and novelty) are still a bit all over the place and cannot convince me it is ready for publication. I would be willing to raise my score, though, should the arguments be refined.
> >
> > —
> >
> > Some additional benefits of VI not covered by the authors that can be used as motivation: Samples from the pre-generator is i.i.d. (whereas MCMC samples are potentially correlated, which might not be ideal). Another useful feature of VI is the fact that it can be amortized, and one can hope to generalize to unseen, future observations to condition on. But this is not explored in this work.
> >
> > —
> >
> > I've read the appendix for the additional discussion on the smoothing parameter. Thanks for the effort. It seems there is some optimization problem for small values of $\sigma$. Perhaps a useful threshold of mean absolute error $\leq 1/256$ can be used in practice, as it is the width of the 8-bit quantization bin. Also, how is the MSE larger than 1 for larger $\sigma$ values? Is the error calculated in the logit space, or is the data rescaled some other way?
> >
> > Thanks for the clarification on the hardness result.

---

> > > ### Author Response · Authors · 2020-11-25
> > > **Response to Additional Questions from R1**
> > >
> > > We appreciate the detailed response from R1.  We have made several updates to the manuscript based on the feedback.  Our response to the additional questions are as follows.
> > >
> > > **1. Clarifying the main contribution of the paper**
> > >
> > > We acknowledge your point and have updated the abstract accordingly -- please take a look. To reiterate, the main contribution of our work is the use of a flow-based pre-generator that works in conjunction with a flow base model, which enables straightforward likelihood-based training via VI. This results in an approximate posterior that is itself a flow model and offers efficient sampling, inversion, and likelihood evaluation, which existing approaches do not provide.
> > >
> > > **2. Meaning of "computational flexibility"**
> > >
> > > By "computational flexibility", we are referring to the operations that can be done efficiently using a flow model, but not with other generative models. These are mentioned in the introduction:
> > >
> > > > "Among them, normalizing flow models ... stand out due to their computational flexibility, as they offer efficient sampling, likelihood evaluation, and inversion."
> > >
> > > For downstream tasks such as lossless compression or uncertainty quantification, having these properties is critical. We believe that this is a key benefit of our approach.  For an example comparison, the actor network in [1] provides fast i.i.d. sampling, but inversion and likelihood evaluation are difficult.
> > >
> > > **3. Additional benefits of VI not covered by the authors**
> > >
> > > We appreciate the constructive suggestion. We had mentioned the i.i.d. guarantee in our earlier response (see answer 3), but it was not added to the manuscript -- it has been added now to the list of contributions at the end of Section 1. The possibility of amortization is mentioned in the conclusion.
> > >
> > >
> > > **4. MSE calculation**
> > >
> > > Thanks for catching this -- the MSE calculation was missing averaging over the pixels and has now been fixed.  We clarify that the error is calculated in the pixel space.

---

### Official Review · AnonReviewer2 · 2020-10-28
**I like the motivation of the problem and the solution. There is also a commendable effort to collect empirical evidence. Some reservations; can benefit from a second review round.**

**Rating:** 6
**Confidence:** 3

**Review:**

### Summary:

This paper uses Variational Inference to query pre-trained flow-models. If the flow variable is $x$, then querries are either conditioned on the part of $x=(x_1,x_2)$, or on a differentiable transformation of $x$. Authors first show that it is not trivial to conduct such queries exactly or approximately for a general class of flow-models. The paper then proposes a framework that affords such querries by working in the latent space--empirical evidence favors the proposed approach over contemporary methods.

### Strength:

The problem is well-motivated. The authors outline several instances where one would want to work with a pre-trained model and use it to query over new data. Further, I appreciate the use of proof that the problem is hard to motivate the solution. I also commend the author's effort to collect empirical evidence for their method. I especially like the "Why Ambient VI fails" explanation and find the contour plot beautiful.


### Concerns:

One primary with paper is the lack of clarity and overload of notation. This is especially true for section 3.

The equations 1 and 2, use the distribution $p_{f\circ \hat{f}}(x_2)$ and  $p_{\hat{f}}(x_2)$. The preceding section uses $p_{f\circ \hat{f}}(x)$ and  $p_{\hat{f}}(x)$ for $x = (x_1, x_2)$; it is possible that the authors are referring to the marginal for $x_2$. However, it is not immediately clear from the discussion. Further, there is reference to $y$ which is undefined till that point. More so, if we are talking about marginals, then this necessiates the need to evaluate these marginals.  The authors offer no discussion on this.

In my understanding of the work, $p_f (x_2| \tilde{x_1} = x_1^*)$ is approximated with $p_f (x| \tilde{x_1} = x_1^*)$. Thereafter, we can use $p_f (x| \tilde{x_1} = x_1^*) \approx p_f (x_2,x_1) p_\sigma  (x_1^*|x_1)$ to calculate the eq 1. However, this still leaves me uncertain about the marginal distribution $p_{f\circ \hat{f}}(x_2)$. I also believe if the above explanation is true, then it is bit of a leap.

However, these concerns vanish for the section with differentiable transformations--here, we do not talk about the partitioning of x, so the expressions are straightforward to evaluate.

The easiest way to convince me would be to offer a clear explanation of the method and to straighten out the notation. It will be great to get an algorithm--like the one in Appendix C--for the partitioning case.

#### Minor concerns and suggestions:

In section 2, the last paragraph, the formulation is not unique to the authors' framework. It is the fundamental idea of VI to use ELBO over KL divergence; however, the current presentation makes it feel that this is a novel observation made by the authors.

Section 1, third paragraph: "VI allows for likelihood evaluation and.."--I believe the term likelihood has been used to refer to $\log q$--this is confusing as the term is often reserved for $\log p(x)$ or $\log p(x|z)$. I will suggest being unambiguous with terminology.

Section 1, fourth paragraph: "Specifically, we use variational inference to
learn a distribution in the latent space ..."--I find this sentence hard to parse. How is a distribution "fed" to the pre-trained model? (this is more of a writing concern than anything else.)

Authors can consider using less left margin for bullet points under the heading "our contributions."

### Update after the rebuttal

I think that the ideas in this paper are interesting and can inspire new uses. All of us agree that the problem presented here was important, and there is a lot of work to be done in this domain. However, after reading the discussion with other reviewers and their reviews, I believe the manuscript can benefit from another review round. Specifically, the authors can benefit from a thorough revision of the claims in the paper. Further, I would encourage authors to at least investigate how naive amortization approaches fair (irrespective of the result, authors will develop a stronger case for their line of work.)

---

> ### Author Response · Authors · 2020-11-19
> **Response to Reviewer 2**
>
> We thank R2 for the thoughtful and positive feedback.  As suggested, we have updated the manuscript to avoid overloading of notations and  improve the clarity and quality of our writing. Below we include our detailed response to the questions raised by R2.
>
> **1. Notational clarification and concerns for marginalization.**
> We believe the main source of confusion is the overloading of $\hat{f}$ in two different contexts. For our method, $\hat{f}: \epsilon \mapsto z$ denotes the invertible mapping for the pre-generator $p_{\hat{f}}$. Thus our variational posterior is the composed model $p_{f \circ \hat{f}}(x)$, and we train $\hat{f}$ such that $p_{f \circ \hat{f}}(x)$ approximates the intractable conditional $p_f(x \mid \tilde{x}_1 = x_1^*)$.
>
> Importantly, since the samples from $p_{f \circ \hat{f}}(x)$ are obtained via $z \sim p_{\hat{f}}, x = f(z)$, we can't efficiently compute the _marginal_ likelihood $p_{f \circ \hat{f}}(x_2)$ due to having to integrate out $x_1$.  This is why we use a modified VI objective that minimizes the KL between the variational and true **joint** distributions, i.e. $D_{\rm KL}(p_{f \circ \hat{f}}(x) \Vert p_f(x \mid \tilde{x}_1 = x_1^*)$. Note that while we cannot compute the marginal likelihood $p_{f \circ \hat{f}}(x_2)$, we can still sample from it by first sampling $x \sim p_{f \circ \hat{f}}(x)$ and only taking the $x_2$ portion. But as noted by R2, this concern vanishes in the more general formulation with a  differentiable transformation $T$.
>
> In the case of Ambient VI, we overloaded the notation and let $\hat{f}: \epsilon \mapsto x$ denote the mapping for the variational posterior such that $p_{\hat{f}}(x)$ directly approximates $p_f(x \mid \tilde{x}_1 = x_1^*)$.  We realize that this notation is confusing and updated the manuscript to use $p_g$ to denote the variational posterior for Ambient VI (reflected in equations 2 and 6).  We also added a pseudocode (Algorithm 2) in Appendix C for the sampling procedure for the partitioned case and the general case.
>
> **2. Is $p_f(x_2 \mid \tilde{x}_1 = x_1^*)$ approximated with $p_f(x \mid \tilde{x}_1 = x_1^*)$?**
> No.  As explained in our response above, we approximate the true conditional joint distribution $p_f(x \mid \tilde{x}_1 = x_1^*)$ with our variational distribution $p_{f \circ \hat{f}}(x)$.
>
> **3. Minor concerns and suggestions**
> Thank you for the suggestions. We have incorporated your suggestions into Sections 1 and 2 for clarity and improved writing.

---

### Decision · Program_Chairs · 2021-01-07
**Final Decision**

**Decision:**

Reject

**Comment:**

This paper proposes a method for conditional inference with arbitrary conditioning by creating composed flows. The paper provides a hardness result for arbitrary conditional queries. Motivated by the fact that conditional inference is hard the paper therefore suggests a novel relaxation where the *conditioning* is relaxed.

There were various concerns from the reviewers regarding notation, comparison algorithms, and how the hardness result motivates the smoothing operation introduced. After careful study of the paper and all the comments I find that I am most concerned about the hardness result and how it motivates the smoothing operation that is done. Novel computational complexity results *as such* are not really in the scope of ICLR. There's nothing wrong with having such a result in a paper, of course, but a paper like this should be evaluated on the basis of the algorithm proposed.

Like R4, I do not follow how this hardness result is meant to motivate the smoothing that's applied. The paper is unambiguous that the goal is to do conditional inference. A hardness result is presented for conditional inference, and so a relaxed surrogate is presented. This has a minor problem that it's not clear the relaxed problem avoids the complexity boundary of the original one. There's a larger problem, though. The hardness result has not been sidestepped! The goal is still to solve conditional inference. The algorithm that's presented is still an approximate algorithm for conditional inference. R4 suggests that other approximation algorithms should be compared to. The authors responded to this point, but I am not able to understand the response. For the same reason, I think it is valid to ask for comparison to other approximate inference algorithms (e.g. without smoothing)

None of the above is to say that the smoothing approach is bad. It may very well be. However, I think that either the existing argument should be clarified or a different argument should be given.

Finally here are two minor points (These weren't raised by reviewers and aren't significant for acceptance of the paper. I'm just bringing them up in case they are useful.)

Is Eq. 3 (proof in Appendix B.1) not just an example of the invariance of the KL-divergence under diffeomorphisms?

Proof in appendix B.2 appears to just a special case of the standard chain rule of KL-divergence (e.g. as covered in Cover and Thomas)